# DAG-Math: Graph-of-Thought Guided Mathematical Reasoning in LLMs

**Yuanhe Zhang**
University of Warwick
yuanhe.zhang@warwick.ac.uk

**Ilja Kuzborskij**[*]
Google DeepMind, United Kingdom
iljak@google.com

**Jason D. Lee**
University of California, Berkeley
jasondlee@berkeley.edu

**Chenlei Leng**
Hong Kong Polytechnic University
chenlei.leng@polyu.edu.hk

**Fanghui Liu**[†]
University of Warwick $\rightarrow$ Shanghai Jiao Tong University
fanghui.liu@{warwick.ac.uk,sjtu.edu.cn}

## Abstract

Large Language Models (LLMs) demonstrate strong performance on mathematical problems when prompted with Chain-of-Thought (CoT), yet it remains unclear whether this success stems from search, rote procedures, or rule-consistent reasoning. To address this, we propose modeling CoT as a certain rule-based stochastic process over directed acyclic graphs (DAGs), where nodes represent intermediate derivation states and edges encode rule applications. Within this framework, we introduce **logical closeness**, a metric that quantifies how well a model's CoT trajectory (i.e., the LLM's final output) adheres to the DAG structure, providing evaluation beyond classical PASS@$k$ metrics. Building on this, we introduce the *DAG-MATH* CoT format and construct a benchmark that guides LLMs to generate CoT trajectories in this format, thereby enabling the evaluation of their reasoning ability under our framework. Across standard math reasoning datasets, our analysis uncovers statistically significant differences in reasoning fidelity among representative LLM families-even when PASS@$k$ is comparable-highlighting gaps between final-answer accuracy and rule-consistent derivation. Our framework provides a balance between free-form CoT and formal proofs systems, offering actionable diagnostics for LLMs reasoning evaluation. Our benchmark and code are available at https://github.com/YuanheZ/DAG-MATH.

## 1 Introduction

Large Language Models (LLMs) have demonstrated promising mathematical reasoning abilities on answer/proof-based problems, e.g., Gemini-2.5 (Gemini Team, 2025) and GPT-5 (OpenAI, 2025), DeepSeek-R1 (DeepSeek Team, 2025). A key strategy underlying these successes is the Chain-of-Thought (CoT) (Nye et al., 2021; Wei et al., 2022; Kojima et al., 2022; Zhang et al., 2022), which encourages models to produce intermediate reasoning steps prior to the final answer.

The black-box nature of CoT in LLMs raises a key challenge: how to rigorously model and evaluate LLMs' mathematical reasoning abilities. Prior works formalize reasoning in terms of search (Shalev-Shwartz & Shashua, 2025), probabilistic inference (Prystawski et al., 2023; Feng et al., 2023), and propositional logic (Merrill & Sabharwal, 2023; Kim & Suzuki, 2024; Yin et al., 2025). Intuitively, LLM reasoning needs to identify all required premises (e.g., facts, constraints), and con-

---

[*]Participated in an advisory capacity only.

[†]Corresponding author. School of Mathematical Sciences, Institute of Natural Sciences and MOE-LSC, Shanghai Jiao Tong University, China. **Declaration:** This work, in collaboration with Google DeepMind, was conducted while Fanghui was at the Department of Computer Science, University of Warwick, UK.

duct correct logical inference from premises to reach the conclusion.[1] These operations must be **exact**, in line with the exact learning requirements in György et al. (2025). To test whether LLMs achieve this through CoT, two elements are crucial:

- **A Rigorous Framework**: It is necessary to establish a principled framework that characterizes the mechanisms by which CoT operates in mathematical problem solving. Such a framework should explicitly account for the roles of premises identification and logical inference. This in turn enables a systematic analysis of reasoning behaviors.

- **An Appropriate Evaluation Metric**: A reliable metric is required to assess whether model outputs reflect authentic reasoning processes rather than the application of heuristic or search-based strategies[2]. An effective metric should hence evaluate not only the final correctness but also the logical coherence of intermediate reasoning steps.

Despite recent progress, existing approaches remain limited in both. In terms of **frameworks**, prior work has examined Boolean-circuit analyses (Li et al., 2024), $k$-parity models (Kim & Suzuki, 2024; Yin et al., 2025), and graph-based representations (Dziri et al., 2023; Mukherjee et al., 2025). Graph-based strategies provide a natural way to model CoT as a discrete, step-level abstractions where nodes correspond to intermediate assertions and edges encode inferential dependencies. However, existing graph-based formulations, whether modeling CoT as deterministic subgraph matching in a directed acyclic graph (DAG) (Dziri et al., 2023), random walks on reversible/recurrent Markov chains (Kim et al., 2025), or trees of linear solution paths (Shalev-Shwartz & Shashua, 2025), neglect the improvement from diverse sampling (Wang et al., 2023) and ability to connect disparate knowledge (Yin & Wang, 2025). As a result, they fail to capture long-range and cross-branch dependencies, as well as the goal-directed, absorbing-state nature of CoT.

In terms of **evaluations**, prior work (Dziri et al., 2023; Joshi et al., 2025; Kim et al., 2025; Xu & Sato, 2025) primarily relies on final-answer metrics such as PASS@$k$ while the entire output for inference is overlooked, which leaves it unclear whether it is solved by logical inference or by search. A promising alternative is to use the LEAN programming language (De Moura et al., 2015; Moura & Ullrich, 2021) to formally verify solutions (Google DeepMind, 2024; Ren et al., 2025; Wang et al., 2025; Lin et al., 2025). While LEAN-based verification ensures logical correctness, it presupposes that problems have been already formalized in a proof-oriented form. For answer-based problems, the formalization into Lean must be carried out in advance, which requires substantial expert effort.

Building on empirical and theoretical insights (Ye et al., 2025; Kim et al., 2025), we address these limitations by formalizing CoT as a rule-based stochastic process on directed acyclic graphs (DAGs). This formalization provides a unified and principled framework for both modeling and evaluation of LLM mathematical reasoning. The main contributions of this work are summarized as below:

- **Framework.** In Section 2, we establish a rigorous graph-based framework that formalizes CoT (i.e., the LLM's entire output) in two phases. Phase 1 constructs a task-specific DAG as the search space for generating CoT trajectories. In Phase 2, the LLM generates CoT trajectories over this DAG under certain stochastic transition rules.

- **Logical Closeness and Metric.** In Section 3, we introduce the notion of *logical closeness*, which evaluates whether an LLM solves a problem by searching over possible choices or by applying rigorous logical inference. This yields a new evaluation metric, the *perfect reasoning rate (PRR)* and the related AUC (area under curve) scores for ranking.

- **Benchmark Construction.** In Section 4, we propose the *DAG-MATH* format based on the above concepts, which makes the logical structure of CoT explicit through DAG representations. Using a three-stage prompting method, we construct a benchmark of 2,894 gold-standard DAG-MATH DAGs. Our graph-level statistical analysis shows that harder problems yield larger, sparser DAGs with higher branching complexity, emphasizing the need for LLMs to decompose tasks, track long dependencies, and recombine results effectively. We have released the benchmark in the **supplemental materials**.

- **Empirical Evaluation.** In Section 5, we employ few-shot prompting to guide LLMs (e.g., Gemini, GPT, Qwen3) to produce DAG-MATH formatted CoT trajectories as the final out-

---

[1] Accurate calculation and symbolic execution are also required, see the discussion in Appendix A.1.

[2] Search-based strategies may yield irrelevant information, undermining solution's consistency. LLMs should be able to summarize the searched/thinking results to ensure the final output logic coherence.

put. We find that search can inflate PASS@1 through exploratory branching, while perfect reasoning ability remains comparable across models. Perfectly reasoning corresponds to easier problems; only-correct-answer CoT reflects modest exploratory overhead; and incorrect CoT typically arises from problems exceeding model capacity, where difficulty stems from branching rather than aggregation.

Our framework provides a "Goldilocks principle" that balances the versatility of natural language with the rigor of LEAN, whose proof terms instantiate our DAG formulation. Moreover, we believe the DAG-MATH framework can lay the foundation for a mathematical definition of reasoning in LLMs (paralleling memorization and generalization in supervised learning), and inform future algorithm design for improved reasoning performance of LLMs.

**Notations:** We denote random variable by a capital letter (e.g., $V$) and its realization by the lower-case letter (e.g., $v$). For shorthand, we write $v_{1:t} = (v_1, v_2, \dots, v_t)$ for $t \geq 1$. We denote a DAG by $\mathcal{G} = (\mathcal{E}, \mathcal{V})$, where $\mathcal{V}$ is the node set and $\mathcal{E}$ is the edge set. For a node $v$, we write $\mathrm{pa}(v)$ as its parent set. We denote the input prompt by $\boldsymbol{X}_{\mathtt{in}} \in \mathcal{P}$, where $\mathcal{P}$ is the power set of the vocabulary.

## 2  A DAG FRAMEWORK FOR STEP-LEVEL CoT

Motivated by empirical observations in Bogdan et al. (2025), we study CoT at the **step** level, rather than the token level. This step-level perspective has been widely considered in recent theoretical analyses (Dziri et al., 2023; Hu et al., 2024; Kim et al., 2025; Shalev-Shwartz & Shashua, 2025), as it better captures intermediate reasoning and the logical structure of solutions. We model step-level CoT in a two-phase workflow as below. Phase 1 defines a task-specific DAG, where Phase 2 samples CoT trajectories over this DAG under certain stochastic process. For better illustration, we take the following problem, adapted from MATH-500 (Hendrycks et al., 2021), as a representative example.

> **Logarithmic Count Problem (LCP)**
>
> For how many integers $k \in [-300, 300]$ does the equation $2\log(x-1) = \log k$ have exactly one real solution $x$?

### 2.1  PHASE 1: TASK-SPECIFIC DAG FOR STEP-LEVEL CoT

**Edges and Nodes in Step-Level CoT:**  For mathematical problems, a CoT step is a natural-language derivation of a new conclusion from prior information. Each step has two components: Edge (Justification): This captures the inference that leads to the step's conclusion. The edge explicitly encodes the logical dependency on the problem statement or on previous steps, making the reasoning chain transparent. Node (Conclusion): The node represents the step's conclusion—the state or value derived from the edge's logic and its parent nodes.

Hence, a single CoT step can be viewed as node/edge decomposition, see an example in Appendix A.2 as well as the nodes/edges in Fig. 1 for the logarithmic count problem. Note that such a decomposition is *non-unique* due to semantic variation, such as synonyms or equivalent phrasings. This ambiguity makes it challenging to develop a precise and principled definition of a CoT step. Consequently, prior work (Dziri et al., 2023; Hu et al., 2024; Kim et al., 2025; Shalev-Shwartz & Shashua, 2025; Bogdan et al., 2025) has typically defined steps heuristically—either as text spans or task-specific annotations—without providing a formal definition. As a first attempt, we present an abstract mathematical formulation, with the technical details deferred to Appendix A.3 since they are not essential for understanding the main text. Within this formulation, although the node/edge decomposition for a single step may still be non-unique, the task-specific DAG introduced later can be made **unique**, provided that each step is restricted to a single conclusion.

**Task-Specific DAG:**  Empirical studies (Ye et al., 2025) demonstrate the existence of a latent directed dependency graph within LLMs, present as soon as a question/prompt is posted, before any output is generated. Formally, given a prompt $\boldsymbol{x}_{\mathtt{in}}$, we define the directed graph as

$$\mathcal{G}(\boldsymbol{x}_{\mathtt{in}}) := (\mathcal{V}(\boldsymbol{x}_{\mathtt{in}}), \mathcal{E}(\boldsymbol{x}_{\mathtt{in}})), \quad \text{where } \mathcal{E}(\boldsymbol{x}_{\mathtt{in}}) \subseteq \mathcal{V}(\boldsymbol{x}_{\mathtt{in}}) \times \mathcal{V}(\boldsymbol{x}_{\mathtt{in}}),$$

where $\mathcal{E}(\boldsymbol{x}_{\mathtt{in}})$ is the set of directed edges and $\mathcal{V}(\boldsymbol{x}_{\mathtt{in}})$ is the set of nodes divided into three classes:

- $\mathcal{V}_{\text{in}}(\boldsymbol{x}_{\text{in}})$ denotes the set of *source* nodes, i.e., nodes formulated solely from the input prompt. In Fig. 1, the source nodes are $v_1, v_2$, and $v_3$.

- $\mathcal{V}_{\text{out}}(\boldsymbol{x}_{\text{in}})$ denotes the set of *sink* nodes, i.e., nodes with only incoming edges and no outgoing edges, corresponding to the final answer(s). The **correct** sink node represents the terminal object that matches the ground-truth answer. In Fig. 1, the sink nodes are $v_{10}$ (correct) and $v_{11}$ (incorrect).

- $\mathcal{V}_{\text{inter}}(\boldsymbol{x}_{\text{in}}) := \mathcal{V}(\boldsymbol{x}_{\text{in}}) \setminus \left( \mathcal{V}_{\text{in}}(\boldsymbol{x}_{\text{in}}) \cup \mathcal{V}_{\text{out}}(\boldsymbol{x}_{\text{in}}) \right)$ denotes the set of intermediate nodes. In Fig. 1, the intermediate nodes are $v_4$ through $v_9$.

We make the following assumption on the acyclic structure of the graph for the absence of circular dependencies, ensuring that no CoT step depends on its own output either directly or indirectly.

**Assumption 1.** *For any input prompt $\boldsymbol{x}_{\text{in}}$, the task-specific directed graph $\mathcal{G}(\boldsymbol{x}_{\text{in}})$ is acyclic.*

If the correct sink node is included in $\mathcal{G}(\boldsymbol{x}_{\text{in}})$, the task-specific DAG can be always **constructed by backtracking** through its ancestors. Note that, the LLM may reason imperfectly during its thinking process, e.g., dead-ends, self-correction, but is expected to output only the finalized, perfect, correct reasoning results to the user (with formal definition later). The reasoning evaluation in this paper is based on the entire final output while PASS@$k$ just considers the final answer. More discussion can be found in Appendix A.4.

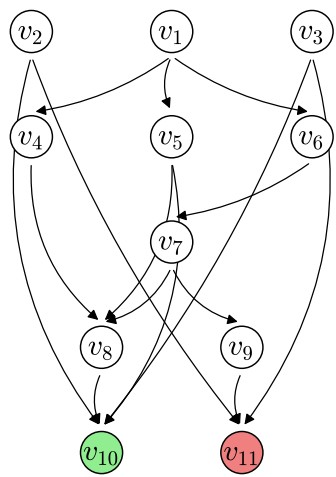

Figure 1: Task-specific DAG via `LLaMA-3.1-8B-Instruct`.

**Task − specific DAG for the logarithmic count problem**

- $v_1$ : "$2\log(x-1) = \log k$" (target equation);
- $v_2$ : "$k \in [-300\,,300]$" (range constraint);
- $v_3$ : "exactly one solution" (task requirement);
- $v_4$ : "$\log(x-1)$ requires $x > 1$" (constraint inferred from the equation);
- $v_5$ : "$\log k$ requires $k > 0$" (constraint inferred from the equation);
- $v_6$ : "$2\log(x-1) = \log k \Rightarrow (x-1)^2 = k$" (re-arranged equation);
- $v_7$ : "$x = 1 \pm \sqrt{k}$" (solve the quadratic equation);
- $v_8$ : "$1 + \sqrt{k}$ is the only solution" (correct check);
- $v_9$ : "For any $k$, there are two solutions" (incorrect check);
- $v_{10}$ : "There are 300 valid values for $k$" (the correct answer);
- $v_{11}$ : "There are 0 valid value for $k$" (the incorrect answer).

## 2.2 PHASE 2: STOCHASTIC PROCESS ON LOGIC DEPENDENCE

Based on the task-specific DAG, the LLM generates CoT trajectories over this DAG as the final output via a certain sampling strategy. Given $\mathcal{G}(\boldsymbol{x}_{\text{in}})$ from Phase 1, we denote the node-level autoregressive distribution of an LLM as $\mathbb{P}$. A node-level CoT trajectory $\{V_i\}_{i=1}^{L}$ with length-$L$, given the input prompt $\boldsymbol{X}_{\text{in}}$, sequentially generates $V_t \in \mathcal{V}$ ($1 \le t \le L$), ultimately leading to the final answer $V_L := V_{\text{out}}$. Specifically, the trajectory $\{V_i\}_{i=1}^{L}$ follows the stochastic process:

$$V_1 \sim \mathbb{P}(\cdot \mid \boldsymbol{X}_{\text{in}}), \cdots, V_t \sim \mathbb{P}(\cdot \mid V_{t-1}, \ldots, V_1, \boldsymbol{X}_{\text{in}}), \cdots, V_{\text{out}} \sim \mathbb{P}(\cdot \mid V_{L-1}, \cdots, V_1, \boldsymbol{X}_{\text{in}}).$$

Next, we define a stochastic transition rule to generate the node-level trajectory over $\mathcal{V}(\boldsymbol{x}_{\text{in}})$. We begin with the initial step, where $\mathbb{P}(V_1 = v \mid \boldsymbol{X}_{\text{in}} = \boldsymbol{x}_{\text{in}})$ is nonzero only for $v \in \mathcal{V}_{\text{in}}(\boldsymbol{x}_{\text{in}})$ and zero for all other nodes. Given $\mathcal{G}(\boldsymbol{x}_{\text{in}})$ and the previous $(t-1)$ steps $V_{1:t-1} = v_{1:t-1}$, the transition probability for the next step is not based on all previous nodes but depends on certain nodes, i.e.

$$\mathbb{P}(V_t = v \mid V_{1:t-1} = v_{1:t-1}, \boldsymbol{X}_{\text{in}} = \boldsymbol{x}_{\text{in}}), \forall v \in \mathcal{V}(v_{1:t-1} \mid \boldsymbol{x}_{\text{in}}),$$

$$\text{with } \mathcal{V}(v_{1:t-1} \mid \boldsymbol{x}_{\text{in}}) := \left\{ v \in \mathcal{V}(\boldsymbol{x}_{\text{in}}) : \text{pa}(v) \subseteq \{v_{1:t-1}\}, v \notin \{v_{1:t-1}\} \right\}, \tag{1}$$

and zero probability for $\forall v \notin \mathcal{V}(v_{1:t-1} \mid \boldsymbol{x}_{\text{in}})$. The sampling process is absorbing upon reaching any node $v \in \mathcal{V}_{\text{out}}(\boldsymbol{x}_{\text{in}})$, indicating that a final answer has been obtained. For **non-thinking LLMs**,

the model directly outputs such types of CoT for the given problem. For **thinking LLMs**, e.g. DeepSeek-R1 (DeepSeek Team, 2025), the thinking process can be viewed as an exploration of the task-specific DAG with self-correction or backtracking, but its final output shown to the users (excluding thinking tokens) is still consistent with our transition rule.

Applied to our logarithmic count problem, Eq. (1) enforces valid transitions over the nodes. For instance, after collecting $\{v_1, v_2, v_3, v_4, v_5\}$, the next admissible node should be $v_6$; while nodes $v_7$ through $v_{11}$ remain inaccessible until $v_6$ has been visited. We use `LLaMA-3.1-8B-Instruct` (Grattafiori et al., 2024) to generate four CoT trajectories, where each trajectory consists of its own steps leading to correct/incorrect answers. The nodes and edges shown in Fig. 1 are performed by the authors, but can also be carried out by LLMs through appropriate prompting (see Section 4). Accordingly, the LLM's final CoT output can be split into three classes:

- **Perfect reasoning** $(v_1, v_2, v_3, v_4, v_5, v_6, v_7, v_8, v_{10})$: The trajectory only includes the correct sink node and its ancestors. We formally define this in the next section.

- **Imperfect reasoning**, e.g., $(v_1, v_2, v_3, v_4, v_5, v_6, v_7, \boxed{v_9}, v_8, v_{10})$: The trajectory still reaches the correct answer but also includes the irrelevant node $v_9$, which is not an ancestor of the correct node. Such case may occur when the LLM explores multiple directions and eventually arrives at the correct answer either by chance or through subsequent derivation. We give a real example for this case in Appendix B where two solution paths are mixed.

- **Wrong reasoning**, e.g., $(v_1, v_2, v_3, v_6, v_7, v_9, v_{11})$: The final answer is incorrect.

**Comparison with previous work:** Our framework integrates an instance-specific DAG with a rule-based stochastic process, directly addressing key limitations in prior work. We do not assume a fixed "universal" graph with deterministic matching Dziri et al. (2023). Instead, our DAG is logical, acyclic, and features absorbing sink nodes, preserving the directional, goal-oriented nature of mathematical problem solving while allowing for long-range dependencies, in contrast to reversible Markov chain models (Kim et al., 2025). Furthermore, unlike the tree abstractions in Shalev-Shwartz & Shashua (2025), our DAG captures shared sub-derivations and the true dependency structure rather than just linear solution paths, thereby supporting multiple valid routes to a solution.

**Universality via Lean 4:** The DAG framework instantiates the same dependency structure that underlies formal proof assistants, indicating that it is not an artifact of natural-language CoT. A Lean 4 proof term is literally a DAG: its leaves are in-scope hypotheses (source nodes in $\mathcal{V}_{\text{in}}$), its internal nodes are applications of previously-proved lemmas or primitive inference rules ($\mathcal{V}_{\text{inter}}$), and its root is the goal proposition (the correct sink in $\mathcal{V}_{\text{out}}$). The Lean kernel type-checks this DAG and rejects any cyclic reference between declarations, realizing Assumption 1 by construction.

## 3 FORMAL DEFINITION OF MATHEMATICAL REASONING ABILITY

Based on our DAG framework, we now present a formal definition of mathematical reasoning ability. Given an input prompt $\boldsymbol{x}_{\text{in}}$, we independently draw $N$ CoT trajectories $\{\boldsymbol{v}^{(i)}\}_{i=1}^{N}$ under the proposed sampling mechanism in Eq. (1). For each trajectory $\boldsymbol{v}^{(i)}$, we construct a trajectory-specific DAG:

$$\mathcal{G}_{\text{gen}}^{(i)}(\boldsymbol{x}_{\text{in}}) = \big(\mathcal{V}_{\text{gen}}^{(i)}(\boldsymbol{x}_{\text{in}}), \mathcal{E}_{\text{gen}}^{(i)}(\boldsymbol{x}_{\text{in}})\big), \quad 1 \le i \le N,$$

where the subscript `gen` indicates that the object (DAG, node, or edge) is extracted from the generated CoT trajectory. Here, $\mathcal{V}_{\text{gen}}$ corresponds to the enumerated steps in the trajectory, and $\mathcal{E}_{\text{gen}}$ contains edges explicitly defined by the parents of each step. Each trajectory-specific DAG is a sub-DAG of $\mathcal{G}(\boldsymbol{x}_{\text{in}})$, and the reasoning ability of each trajectory can be evaluated using a new metric, termed **logical closeness**, and the concept of *perfect reasoning*, introduced in our framework.

**Definition 1** (Logical closeness and perfect reasoning). *Under Assumption 1, consider an input prompt $\boldsymbol{x}_{\text{in}}$ and the DAG $\mathcal{G}_{gen}(\boldsymbol{x}_{\text{in}})$. For each node $v \in \mathcal{G}_{gen}(\boldsymbol{x}_{\text{in}})$, define its out-degree as*

$$\deg\big(v \mid \mathcal{G}_{gen}(\boldsymbol{x}_{\text{in}})\big) := \Big|\big\{u \in \mathcal{G}_{gen}(\boldsymbol{x}_{\text{in}}) \mid (v \to u) \in \mathcal{E}_{gen}(\boldsymbol{x}_{\text{in}})\big\}\Big|.$$

*We say that $\mathcal{G}_{gen}(\boldsymbol{x}_{\text{in}})$ is **logically closed** if*

$$\deg\big(v \mid \mathcal{G}_{gen}(\boldsymbol{x}_{\text{in}})\big) \ge 1, \quad \forall v \in \mathcal{V}_{gen}(\boldsymbol{x}_{\text{in}}),$$

*i.e., only the final nodes have no outgoing edges. Furthermore, if the sink node corresponds to the correct answer, we call the associated CoT trajectory a case of **perfect reasoning**.*

Any topological ordering of the ancestor nodes that terminates at the correct sink node is perfect reasoning. Compared with evaluating reasoning based solely on the correctness of the final answer, incorporating logical closeness allows us to assess whether an LLM engages in genuine logical inference rather than merely searching among possible solutions. Based on the definition of logical closeness, we now formally define the mathematical reasoning ability of LLMs as follows.

**Definition 2** (Mathematical reasoning ability). *Under Assumption 1, let an LLM be given a prompt $\boldsymbol{X}_{\text{in}} \in \mathcal{P}$, sampled from an underlying distribution $\mathcal{D}$ over mathematical problem prompts. We define two indicator functions for a trajectory-specific DAG $\mathcal{G}_{gen}(\boldsymbol{X}_{\text{in}})$:*

$$\delta_{\text{close}}\big(\mathcal{G}_{gen}(\boldsymbol{X}_{\text{in}})\big) := \begin{cases} 1, & \text{if } \mathcal{G}_{gen}(\boldsymbol{X}_{\text{in}}) \text{ is logically closed,} \\ 0, & \text{otherwise,} \end{cases}$$

$$\delta_{\text{final}}\big(\mathcal{G}_{gen}(\boldsymbol{X}_{\text{in}})\big) := \begin{cases} 1, & \text{if the sink node of } \mathcal{G}_{gen}(\boldsymbol{X}_{\text{in}}) \text{ is correct,} \\ 0, & \text{otherwise.} \end{cases}$$

*Then, the **Perfect Reasoning Rate (PRR)** of an LLM w.r.t. a given prompt $\boldsymbol{X}_{\text{in}}$ is defined as*

$$\text{PRR}(\boldsymbol{X}_{\text{in}}) := \mathbb{E}_{\mathbb{P}}\Big[\delta_{\text{close}}\big(\mathcal{G}_{gen}(\boldsymbol{X}_{\text{in}})\big) \times \delta_{\text{final}}\big(\mathcal{G}_{gen}(\boldsymbol{X}_{\text{in}})\big)\Big].$$

*The overall **mathematical reasoning ability** of an LLM over the distribution $\mathcal{D}$ is then measured as*

$$\mathcal{R} := \mathbb{E}_{\boldsymbol{X}_{\text{in}}\sim\mathcal{D}}\big[\text{PRR}(\boldsymbol{X}_{\text{in}})\big] = \mathbb{E}_{\mathbb{P},\,\boldsymbol{X}_{\text{in}}\sim\mathcal{D}}\Big[\delta_{\text{close}}\big(\mathcal{G}_{gen}(\boldsymbol{X}_{\text{in}})\big) \times \delta_{\text{final}}\big(\mathcal{G}_{gen}(\boldsymbol{X}_{\text{in}})\big)\Big].$$

**AUC socres:** By relaxing $\delta_{\text{close}}$ to permit a certain proportion of nodes that do not satisfy logical closeness, we obtain the corresponding AUC scores (with proportion of logic closeness from 0% to 100%), which serves as a comprehensive measure of mathematical reasoning performance.

This metric can be regarded as a combination of the final-answer correctness reflected by PASS@1 and logical closeness. To illustrate Definition 2, consider a toy DAG example in Fig. 2, consisting of two linear chains of length $L$ emanating from a common source node. We denote the correct sink node as $\textsf{L}$ and the incorrect sink node as $\textsf{L}'$. At each step, the transition distribution $\mathbb{P}$ is uniform over all available nodes according to Eq. (1), i.e.,

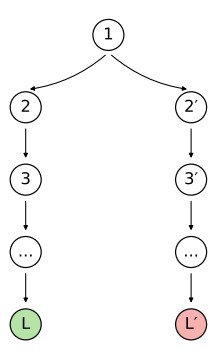

$$\forall\, v \in \mathcal{V}(v_{1:t-1} \mid \boldsymbol{x}_{\text{in}}), \quad \mathbb{P}(V_t = v \mid V_{1:t-1} = v_{1:t-1}, \boldsymbol{X}_{\text{in}} = \boldsymbol{x}_{\text{in}}) = \frac{1}{2}.$$

To remain logically closed, a trajectory must stay on the same chain across the remaining $L - 1$ transitions, each occurring with probability $1/2$. Hence, we have $\text{PRR} = \left(\frac{1}{2}\right)^{L-1}$. This illustrates that PRR decays **exponentially with depth**: although the final-answer accuracy $(1/2)$ may appear stable, logically closed trajectories become increasingly rare. Furthermore, Appendix B.1 presents a geometry example from AIME 2025 (Art of Problem Solving, 2025a) where the final answer is correct, but logic-closeness fails due to mix-

Figure 2: Toy DAG.

ing two solution paths. Consequently, lacking logic closeness risks producing answers correct only by chance, obscuring flawed reasoning, reducing reliability, and undermining interpretability. Furthermore, our framework also can be extended to analyze IMO-level problems via lemma-level DAG, which can reveal the key structural distinction. A concrete example based on the problem 4 from IMO 2025 (IMO Foundation, 2025) is presented in Appendix B.2.

In practice, given a dataset with $M$ problems and $N$ independent CoT trajectories per problem, one can approximate $\text{PRR}(\boldsymbol{x}_{\text{in}})$ and $\mathcal{R}$ by

$$\widehat{\text{PRR}}(\boldsymbol{x}_{\text{in}}) := \frac{1}{N}\sum_{i=1}^{N} \delta_{\text{close}}^{(i)}(\boldsymbol{x}_{\text{in}}) \times \delta_{\text{final}}^{(i)}(\boldsymbol{x}_{\text{in}}), \quad \widehat{\mathcal{R}} := \frac{1}{M}\sum_{j=1}^{M} \widehat{\text{PRR}}\left(\boldsymbol{x}_{\text{in}}^{(j)}\right).$$

By Hoeffding's inequality, taking $M = \Omega(1/\varepsilon^2)$ is sufficient for $\left|\widehat{\mathcal{R}} - \mathcal{R}\right| \leq \varepsilon$ with high probability.

**Comparison with generalization in supervised learning:** Analogous to underfitting and overfitting in supervised learning, we can define *under-reasoning*—where the DAG omits necessary

intermediate steps-and *over-reasoning*-where the DAG is logically sound but contains redundant or irrelevant steps. In both cases, the estimated reasoning ability $\widehat{\mathcal{R}}$ is low, and *perfect reasoning* can be regarded as the "sweet spot" in this reasoning-generalization analogy.

Importantly, $\mathcal{R}$ offers a richer error taxonomy, distinguishing structurally illegal paths, legal-but-wrong paths, and "imperfectly correct" trajectories, whereas standard generalization theory typically treats all errors uniformly. Regularization strategies, such as the minimum description length principle (Grünwald, 2007), could potentially mitigate over-reasoning by favoring concise proofs that conform to a proof grammar or template. We leave exploration of this direction to future work.

# 4 DAG-MATH FORMATTED CoT AND BENCHMARK

In practice, standard CoT trajectories generated by LLMs are unstructured, autoregressive sequences of tokens. Within such free-form text, the logical steps (nodes) and their dependencies (edges) are often entangled, which complicates the evaluation of our step-level reasoning framework. To address this issue, we introduce a structured CoT format via prompting, described in Section 4.1, which we term the *DAG-MATH* format. This format facilitates the construction of the corresponding DAG, enabling the creation of a DAG benchmark, presented in Section 4.2.

## 4.1 DAG-MATH FORMATTED CoT

As a structured, step-by-step format, *DAG-MATH* explicitly specifies each reasoning step in forward generation order: *Edge → Parent(s) → Node*. This ordering is designed for evaluation-oriented sampling: first, declare the logical link to prior knowledge (`Edge`); next, cite the necessary antecedents (`Parents`); and finally, assert the derived conclusion (`Node`). The DAG-MATH format can be produced by LLMs through prompt engineering. For illustration, below is one step from the DAG-MATH formatted CoT for the logarithmic count problem (all steps in DAG-MATH format are presented in Appendix C).

---

**Step 4**

**Edge**: Since the left-hand side of the equation in Step 1 contains $\log(x-1)$, the domain restriction for a logarithm requires $x - 1 > 0$, i.e., $x > 1$.
**Parents**: Step 1.
**Node**: $\log(x-1)$ requires $x > 1$.

---

Step IDs also serve as node identifiers, which allows for a straightforward evaluation of DAG closeness. Based on the DAG-MATH format, a *gold-standard* CoT trajectory is defined by three criteria: (1) it adheres to the DAG-MATH format; (2) its corresponding DAG is logically closed; and (3) the final answer is correct.

## 4.2 GOLD-STANDARD DAG-MATH BENCHMARK

We prompt LLMs to generate CoT trajectories in the DAG-MATH format for existing mathematical datasets, such as Omni-MATH (Gao et al., 2024), and construct the corresponding DAGs. By verifying both logical closeness (Definition 1) and the correctness of the final answers, we compile a benchmark consisting of 2,894 gold-standard DAGs. The primary **purpose of this benchmark** is to characterize the statistical properties of these gold-standard DAGs across different problem difficulty levels, providing valuable insights for evaluating and enhancing LLM mathematical reasoning.

The benchmark comprises problems from Omni-MATH (Gao et al., 2024), which are categorized into difficulty levels ranging from 1 (easiest) to 10 (hardest). To ensure high solvability by LLMs, we only consider problems with difficulty levels below 6. For generating DAG-MATH formatted CoTs, we employ `GPT-o4-mini` and `Qwen3-235B-A22B-Thinking-2507`, both recognized as leading models in mathematical problem solving (LMArena, 2025). Gold-standard CoTs are constructed using a three-stage prompting strategy (see Appendix D) in **reverse order** (Node → Parents → Edge). This approach fixes the node set first, making verification easier with SymPy or LLM-as-Judge and minimizing error propagation, thereby ensuring high-quality trajectories. Furthermore, we perform human evaluation on 50 random samples and the results (49/50) confirm the reliability of our benchmark, see more details in Appendix E.

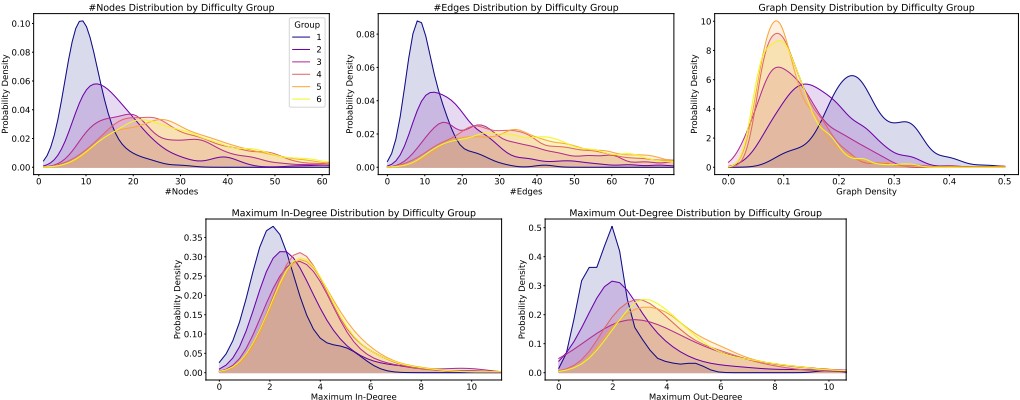

Figure 3: Empirical distributions of DAG statistics across problem difficulty groups, where group $k$ corresponds to problems with difficulty in $(k-1, k]$. Shown are the distributions of #Nodes, #Edges, graph density $\left(\text{i.e., } \frac{2\#\text{Edges}}{\#\text{Nodes}(\#\text{Nodes}-1)}\right)$, maximum in-degree, and maximum out-degree.

We consider five representative **graph-level statistics**: (1) the total number of nodes (#Nodes); (2) the total number of edges (#Edges); (3) graph density, defined as the ratio of #Edges to the maximum possible number of edges in an acyclic graph, i.e., $\frac{2\#\text{Edges}}{\#\text{Nodes}(\#\text{Nodes}-1)}$; (4) the maximum in-degree, denoted $\text{d}_{\text{in}}^{\text{max}}$; and (5) the maximum out-degree, denoted $\text{d}_{\text{out}}^{\text{max}}$. Fig. 3 shows the distributions of these five statistics across problem difficulty levels. Our key observations as **problem difficulty increases from 0 to 6** are as follows:

- **More nodes and edges with heavier tails:** The distributions of #Nodes and #Edges shift noticeably to the right and develop heavier tails, indicating that harder problems produce larger graphs, while simpler problems yield much smaller ones.

- **Sparser structure:** The graph becomes sparse when problem difficulty increases. Harder reasoning produces broader, less connected structures, reflecting modular sub-reasoning where semi-independent chains (e.g., sub-tasks or lemmas) are later combined.

- **Logic complexity reflected in maximum out-degree:** As difficulty increases, the distributions of maximum in-degree and out-degree shift rightward with heavier tails. Maximum in-degree grows slowly, suggesting most steps rely on few inputs, whereas maximum out-degree rises more sharply, indicating that certain key steps support multiple inferences. This implies that logical complexity scales primarily through branching rather than aggregation. The average in- and out-degree remains around 1.3 across difficulty levels, as most nodes have small degrees while a few pivotal steps exhibit large connectivity.

Accordingly, as problems become harder, their DAGs grow larger and sparser, with complexity arising primarily from branching into modular sub-reasoning chains. This underscores the importance of LLMs being able to decompose problems into sub-tasks, track longer dependencies, and recombine intermediate results to solve challenging problems effectively.

## 5 EVALUATION OF MATHEMATICAL REASONING ABILITY

To evaluate mathematical reasoning performance, we employ few-shot prompting (using demonstrations from the benchmark in Section 4.2; see Appendix F for details) to guide LLMs in generating DAG-MATH formatted CoTs/DAGs for test problems, without providing the final solutions.

**Models and datasets:** We evaluate five LLMs: `Gemini-2.5-Flash` (Gemini-2.5-F), `Gemini-2.5-Flash-Lite` (Gemini-2.5-F-L), `GPT-4.1`, `GPT-4.1-mini` (GPT-4.1-M), and `Qwen3-30B-A3B-Instruct-2507` (Qwen3-30B). These models demonstrate strong mathematical performance even without long thinking (White et al., 2025), also are more efficient and economical than other thinking-focused models due to lower token usage. We evaluate these models

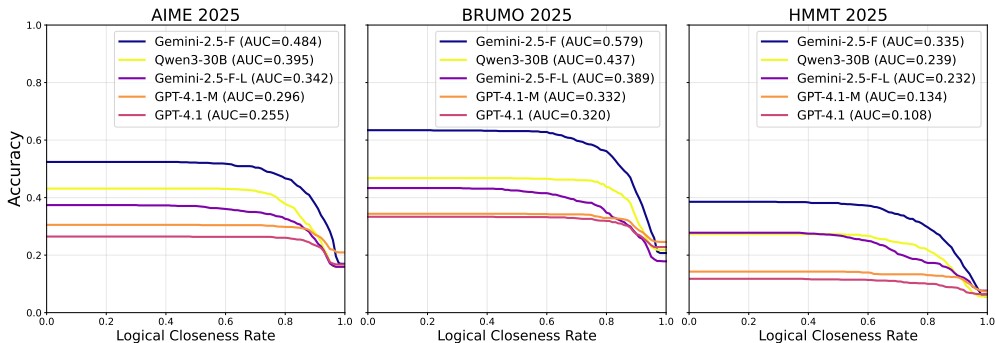

Figure 4: The AUC curves of averaged accuracy under the logical closeness rate over three datasets for five selected LLMs. The curve is a direct plot of the computed accuracies at a fine-grained grids of threshold (0.01 increment from 0 to 1).

on three recently adopted datasets for high-difficulty, answer-based problems: AIME 2025 (Art of Problem Solving, 2025a;b), BRUMO 2025 (BRUMO, 2025), and HMMT 2025 (HMMT, 2025).

**Experimental Settings:** For each model, we use a 4-shot prompting strategy to generate 32 DAG-MATH formatted CoT trajectories per problem across all datasets. Next, we extract the corresponding DAGs and evaluate the five graph-level statistics introduced in Section 4.2, as well as model performance metrics (e.g., PASS@1 and $\widehat{\mathcal{R}}$). The sampled DAGs are then partitioned into four classes: **All** (no filtering), **Incorrect** (ending at an incorrect sink), **Correct** (ending at the correct sink), and **Perfect** (logically closed and ending at the correct sink).

## 5.1 RESULTS

Fig. 4 reports AUC scores across three datasets, with PASS@1 as the starting point and $\widehat{\mathcal{R}}$ as the end point (see more details in Table 3 in Appendix I), as the logic closeness rate increases. Besides, we also report averaged graph-level statistics for AIME 2025 in Table 4, with additional results for BRUMO 2025 and HMMT 2025 in Appendix I. We have the following observations:

- **Search improves raw accuracy while perfect reasoning ability remains similar.** All models exhibit a noticeable drop from PASS@1 to $\widehat{\mathcal{R}}$; while PASS@1 varies widely across models, $\widehat{\mathcal{R}}$ remains relatively stable (i.e., the end point is almost the same). This suggests that additional exploration or search can inflate raw accuracy, while the models' inherent perfect reasoning ability is broadly comparable. We conduct a statistical test in Appendix K.1 to show statistical significance of those gaps. The AUC scores indicate that models with correct answers achieve at most 80% logic-closed nodes, but accuracy degrades markedly under stricter criteria. This suggests that outputs, while correct, are superficially consistent at some point, which aligns with users' impressions when using these LLMs.

- **Graph structure reflects problem difficulty and reasoning quality.** Harder problems induce larger, sparser, and more branchy DAGs (see Section 4.2). Within each model, **Perfect** reasoning trajectories correspond to the smallest, densest graphs, reflecting concentrated reasoning on solvable items. **Correct** graphs are slightly larger and sparser, suggesting the inclusion of useful exploratory steps. In contrast, **Incorrect** graphs exhibit strong branching (with $\widehat{d}_{\text{out}}^{\max}$ growing faster than $\widehat{d}_{\text{in}}^{\max}$), indicating that failure often arises from speculative expansions rather than from aggregating insufficient inputs. For example, Gemini's **Correct** cohort exhibits larger but sparser graphs with slightly higher branching when compared to the GPT family, indicating that effective exploration and task planning can increase the likelihood of reaching correct answers without fully closed, deeper reasoning.

- **Identifying the "difficulty boundary".** Within each model, the **Incorrect** cohorts resemble "harder-than-ability" graphs (see Section 4.2): larger, sparser, heavy-tailed, and with notably higher $\widehat{d}_{\text{out}}^{\max}$. In contrast, the **Perfect** cohorts converge to smaller, relatively dense DAGs with low branching. This indicates that each model's effective difficulty ceiling corresponds to the regime where it can maintain compact, low-branching DAGs; beyond this point—when branching explodes and density drops—accuracy sharply declines.

These insightful findings from our proposed metrics can provide a principled foundation for future work aimed at improvement, see a detailed discussion in Appendix J.

## 5.2 MORE EXPERIMENTS AND DISCUSSIONS

Here we discuss several additional experiments to validate our claims and findings.

**Reasoning Gap:** We expand evaluation sample size from 32 to 128 trajectories per problem and calculate 95% confidence intervals for the gap between PASS@1 and PRR. The resulting intervals were tight and consistently excluded zero (e.g., confirming a 36% gap for `Gemini-2.5-Flash`), statistically demonstrating that the difference between finding the correct answer and generating a perfectly valid derivation is significant and substantial across all tested models. More details are presented in Appendix K.1.

**Prompt Sensitivity:** We employ Two One-Sided Tests (TOST) to verify that DAG-MATH evaluations are robust to variations in prompt. By perturbing the few-shot prompts through re-formatting (changing nested bullets to tables) and re-phrasing (using synonyms and alternative grammar), then generate new sets of DAGs. The analysis shows statistical equivalence across five key graph-level statistics and performance metrics, see more details in Appendix K.2.

**Parsing Consistency and Non-Circularity:** We validate the objectivity of PRR/AUC metric by examining the consistency of DAG construction across different parsing methods, including self-parsing and external parsing by Thinking LLMs. The AUC scores for logical closeness remained remarkably stable across all models (ranging from 23.8% to 25.5%) with statistically equivalent DAG structures tested by TOST, demonstrating that logical closeness is an intrinsic property of the solution rather than an artifact of the parser. More details are reported in Appendix K.3.

**Formatting Constraints:** By employing external models to parse natural CoT into dependency graphs, the analysis revealed that the metrics effectively capture fundamental reasoning patterns, with Natural CoT achieving AUC scores comparable to the structured format and over 70% logical closeness. This confirms that the evaluation framework is not limited to the benchmark but serves as a robust, objective method for diagnosing the logical coherence and reasoning fidelity of freeform LLM generations. More details are presented in Appendix K.3.

**Cross-Family Few-Shot Demonstrations:** We investigate potential biases introduced by the model source of few-shot demonstrations. While using cross-family examples resulted in a slight performance decrease (`GPT-4.1` PRR drops from 16.8% to 15.9%), the DAG structures are statistically equivalent and the relative rankings of the models remained unchanged. This indicates that while same-family demonstrations are optimal, our framework's evaluations remain valid even when cross-family prompting is employed. More details are presented in Appendix K.4.

**Role of Thinking:** We perform experiments with thinking modes for two Gemini models om AIME 2025. The results in Table 1 confirm that while thinking mode significantly boosts both PASS@1 and PRR, the substantial performance gap between two metrics persist. The continued gap indicates that thinking enhances the model's exploration of the task-specific DAG, yielding more correct and logically closed trajectories, but does not eliminate the tendency on search over logical coherence.

Table 1: Results of PASS@1 and PRR for Gemini models w/wo thinking on AIME 2025.

|  | PASS@1 | | PRR | |
|---|---|---|---|---|
| **Thinking** | w/o | w | w/o | w |
| Flash | 52.4% | 64.0% | 17.0% | 35.9% |
| Flash-Lite | 37.4% | 49.2% | 15.9% | 24.5% |

## 6 CONCLUSION AND CONJECTURE

This paper proposes a novel DAG-MATH framework for modeling and evaluating mathematical reasoning, introducing the concepts of logical closeness and perfect reasoning over DAGs. We demonstrate how DAG graph statistics vary with problem difficulty and how models' perfect-reasoning ability and AUC curve behaves across these tasks. We conjecture that every "perfect" final output of an LLM for reasoning-intensive tasks, such as mathematical proofs, derivations, or structured diagnostic report, admits a DAG representation at an appropriate semantic level, where nodes correspond to atomic claims or intermediate conclusions and edges encode their dependency relations.

## ACKNOWLEDGMENT

Y. Z. was supported by Warwick Chancellor's International Scholarship. JDL acknowledges support of Open Philanthropy, NSF IIS 2107304, NSF CCF 2212262, ONR Young Investigator Award, NSF CAREER Award 2144994, and NSF CCF 2019844. F. L. was supported by Royal Society KTP R1 241011 Kan Tong Po Visiting Fellowships and Warwick-SJTU seed fund. We thank Zulip[3] for the project organization tool and Sulis[4] for GPU computation resources.

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

CONTENTS

# A    ILLUSTRATION AND DEFINITION OF COT STEP

In this section, we first discuss execution correctness as a measure of LLM reasoning performance and emphasize that our framework addresses the key challenge of tracking logical dependencies. To facilitate understanding, we first provide an intuitive illustration of a CoT step using the logarithmic count problem, followed by a formal mathematical definition of a CoT step.

## A.1    REMARK ON EXECUTION CORRECTNESS

We note that accurate calculation and symbolic execution remain essential for evaluating an LLM's mathematical reasoning. In practice, LLMs may make stepwise errors, such as computational slips or misreading problem statements. Our framework, however, assumes that each step is correct. This simplification is justified because step-level errors can be automatically detected using Process Reward Models (Lightman et al., 2023; Wang et al., 2024; Zhang et al., 2025) or SymPy validation.

Focusing on correct steps allows us to capture the trajectory of model capabilities: with recent advances (Kavukcuoglu, 2025; OpenAI, 2025; Yang et al., 2025; DeepSeek Team, 2025), the main bottleneck in complex mathematical reasoning lies less in local step fidelity and more in the higher-level logical structure. Our framework addresses this challenge by: (1) perceiving the full logical structure of a problem, and (2) navigating it to construct a coherent solution path. By abstracting away from local step errors, we can isolate and analyze the structural core of CoT reasoning.

## A.2    INTUITIVE UNDERSTANDING OF STEPS IN COT

We illustrate a single step in the logarithmic count problem (LCP) below.

> **One CoT Step in the Logarithmic Count Problem (LCP)**
>
> If two logarithmic expressions are equal, then their arguments must also be equal.
>
> Hence, from $2\log(x-1) = \log k$ , we can conclude $(x-1)^2 = k$ .

In this illustration, the blue part corresponds to the previous conclusion (parent node), the green part represents the new conclusion for the current node, and the orange part highlights the logical reasoning (Edge) that connects the parent to the current node. Note that the edge may be latent when the model only outputs the conclusion without explicitly stating the reasoning.

## A.3    MATHEMATICAL DEFINITION OF STEPS IN COT

We now formalize the definition of a single CoT step. Let $\mathcal{V}$ denote the semantic domain of mathematical objects in *semantic normal form* (SNF). A raw CoT step is a sequence of tokens, i.e., a string $c \in \mathcal{P}$. We introduce a *canonicalization map*

$$\kappa : \mathcal{P} \to \mathcal{V}$$

that maps any textual span to its SNF representation:

$$c \xrightarrow{\ \kappa\ } \kappa(c) \in \mathcal{V}.$$

The canonicalization map satisfies the following properties:

- **Idempotent:** $\kappa(\kappa(c)) = \kappa(c)$.
- **Presentation-invariant:** It does not perform substantive algebraic manipulations (e.g., expanding or factoring), which are treated as separate reasoning steps.

Canonicalization removes superficial variations such as synonyms, spacing differences, commutativity, and $\alpha$-equivalent variable names, ensuring semantic consistency.

Intuitively, a CoT step can be mapped to a normalized SymPy object that captures its underlying mathematical semantics. A concrete example of this canonicalization is provided below.

- *Input:* Raw CoT step as text tokens (e.g., "The value of our target is $x + y$." or "The target value is $(y + x)$.").

- *Canonicalization ($\kappa$):* Maps the input to a normalized form by removing superficial variations such as synonyms, spacing, commutativity, and $\alpha$-equivalent variable names.

- *Output:* A semantic object (e.g., SymPy object Add(x,y)) that consistently encodes the meaning.

Accordingly, let $\mathcal{F}$ denote a signature of *primitive inference rules/operations*. Each $f \in \mathcal{F}$ is associated with an arity $\mathrm{ar}(f) \in \mathbb{N}$ (i.e., the number of inputs the rule or operation takes) and a (partial) semantic operator $[\![f]\!] : \mathcal{V}^{\mathrm{ar}(f)} \rightharpoonup \mathcal{V}$. Intuitively, $\mathcal{F}$ represents the atomic reasoning steps allowed at the CoT level, such as a single algebraic operation, one application of a named lemma, or a single substitution. We now formally define a CoT step.

**Definition 3** (Atomic Step of CoT). *Given an input prompt $\boldsymbol{x}_{\mathrm{in}} \in \mathcal{P}$, a CoT trajectory of length $\ell$ is a sequence of steps $\mathcal{C} = (\mathbf{c}_1, \ldots, \mathbf{c}_\ell)$. Suppose $\kappa : \mathcal{P} \to \mathcal{V}$ is a canonicalization map. Then, a step $\mathbf{c}_i$ can be formulated as a triple $(\Gamma_i, f_i, v_i)$ denoted by $\Gamma_i \xrightarrow{f_i} v_i$, where:*

1. ***Canonicalization:*** *Each string $\mathbf{c}_i \in \mathcal{P}$ produced in the CoT trajectory is mapped by $\kappa$ to the corresponding SNF object $v_i = \kappa(\mathbf{c}_i) \in \mathcal{V}$.*

2. ***Premises:*** *$\Gamma_i \subseteq \{v_1, v_2, \ldots, v_{i-1}\}$ is the finite set of previously established SNF objects (from the prompt or earlier steps) directly used to infer the current step.*

3. ***Primitive operation:*** *$f_i \in \mathcal{F}$ and $v_i = [\![f_i]\!](\Gamma_i)$, i.e., $v_i$ is obtained by exactly one application of a primitive operator to the premises.*

Accordingly, each step in a CoT can be viewed as the reasoning pattern:

$$\text{(Premises used)} + \text{(inference rule applied)} \longrightarrow \text{(new result)}.$$

Next, we provide a concrete algebra example on expanding $(x + y)^2$ for better intuition.

**Step $c_i$: Expand the square**

$$\Gamma_i = \{(x + y)^2\}, \quad f_i = \text{"expand square"}, \quad v_i = [\![f_i]\!](\Gamma_i) = (x + y)(x + y).$$

So we have:

$$\{(x + y)^2\} \xrightarrow{\text{expand square}} (x + y)(x + y).$$

**Step $c_{i+1}$: Distribute the product**

$\Gamma_{i+1} = \{(x+y)(x+y)\}, \quad f_{i+1} = \text{"distributive law"}, \quad v_{i+1} = [\![f_{i+1}]\!](\Gamma_{i+1}) = x^2 + xy + yx + y^2.$

So we have:

$$\{(x + y)(x + y)\} \xrightarrow{\text{distribute}} x^2 + xy + yx + y^2.$$

**Step $c_{i+2}$: Simplify like terms**

$$\Gamma_{i+2} = \{x^2 + xy + yx + y^2\},$$
$$f_{i+2} = \text{"commutativity + combine like terms"},$$
$$v_{i+2} = [\![f_{i+2}]\!](\Gamma_{i+2}) = x^2 + 2xy + y^2.$$

So we have:

$$\{x^2 + xy + yx + y^2\} \xrightarrow{\text{simplify}} x^2 + 2xy + y^2.$$

Combining the above steps, the CoT trajectory is

$$(x + y)^2 \xrightarrow{\text{expand square}} (x + y)(x + y) \xrightarrow{\text{distribute}} x^2 + xy + yx + y^2 \xrightarrow{\text{simplify}} x^2 + 2xy + y^2.$$

This example follows Definition 3 and precisely characterizes a CoT trajectory at the step level. Note that this step-level formalization is not essential for understanding the main text, which primarily focuses on DAG-level reasoning rather than the specifics of individual nodes and edges. Nonetheless, for readers interested in how nodes and edges are defined or how they influence a CoT trajectory, this definition and the accompanying example provide a useful reference.

## A.4    DISCUSSIONS ON ACYCLICITY

The acyclic assumption in Assumption 1 is sufficient to characterize both problem-solving and theorem proving. A mathematical derivation/proof is a finite sequence of well-formed formulas where each formula is either an axiom or is derived from earlier formulas by a rule of inference. If we treat each formula occurrence as a node and draw a directed edge from each premise inferred from it, the dependency graph we obtained is indeed a directed acyclic graph. This is also supported by formal finitary system such as Zermelo–Fraenkel set theory or type theory. Strategies like proof-by-contradiction are rules of inference (e.g., $\neg P \rightarrow \bot \vdash P$), which creates a clear, acyclic dependency: the conclusion $P$ depends on the sub-derivation that starts with the assumption $\neg P$ and ends with a contradiction $\bot$. This is not a cycle but a conditional branch that is closed, which can be covered by our assumption.

## B    EXAMPLES OF LOGICAL CLOSENESS

### B.1    A HIGH-SCHOOL COMPETITION-LEVEL EXAMPLE DAG

There are several reasons why LLMs may generate unclosed nodes even though the final answer is correct:

- Assertions stemming from an alternative strategy that is not the one leading to the final answer in the trajectory.

- Qualitative axioms that are implicitly used. When forming edges, the model tends to link parents that provide numerical values from earlier calculations, since quantitative conclusions are easier to cite than qualitative ones.

- Irrelevant information drawn from the problem statement.

- Additional commentary based on previous conclusions but not required for the solution.

We aim to analyze specific DAG-MATH formatted CoT trajectory which has the correct final answer but unclosed DAG. To justify the rationale, we take the following geometry problem from AIME 2025 I (Art of Problem Solving, 2025a) as an example.

> **Area of Heptagon Problem**
>
> On $\triangle ABC$, points $A, D, E$, and $B$ lie in that order on side $\overline{AB}$ with $AD = 4$, $DE = 16$, $EB = 8$. Points $A, F, G$ and $C$ lie in that order on side $\overline{AC}$ with $AF = 13$, $FG = 52$, and $GC = 26$. Let $M$ be the reflection of $D$ through $F$, and let $N$ be the reflection of $G$ through $E$. Quadrilateral $DEGF$ has area 288. Find the area of heptagon $AFNBCEM$.

We have 4 correct CoT trajectories over 32 total trajectories generated by `Gemini-2.5-Flash`. We provide a detailed analysis of one trajectory that contains multiple unclosed patterns and has a moderate graph size. There are two trajectories that exhibit similar characteristics, while the last trajectory consists of 121 nodes.

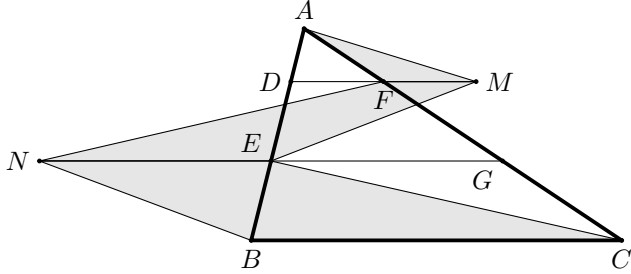

Figure 5: Visualization of the heptagon problem.

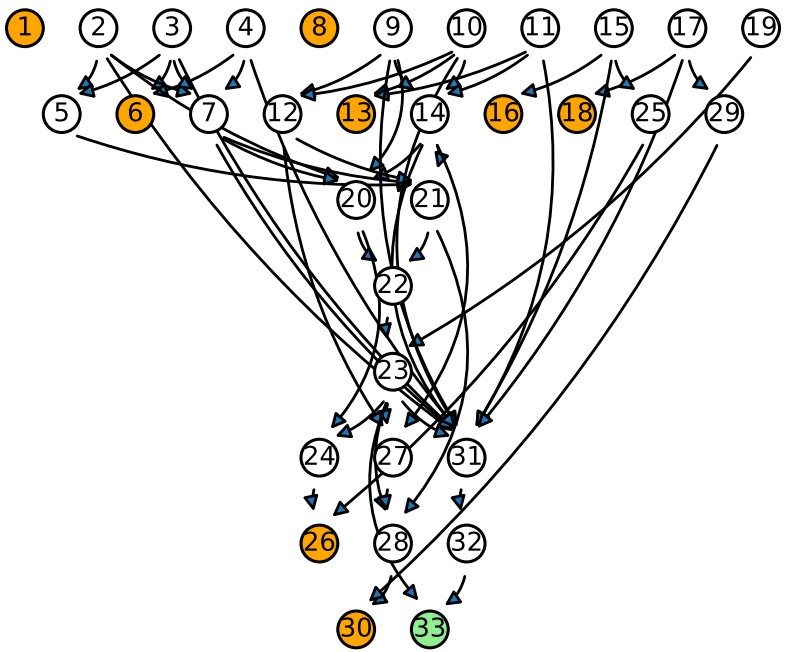

Figure 6: A trajectory-specific DAG for the Area of Heptagon problem, whose DAG-MATH–formatted CoT trajectory is generated by `Gemini-2.5-Flash` and has the correct final node. Nodes without any children are highlighted in orange.

We plot Fig. 5 for better illustration and present the first trajectory-specific DAG shown in Fig. 6. We summarize each node's conclusion below.

1. States the ordering/collinearity $A$, $D$, $E$, $B$ on $\overline{AB}$.
2. States $AD = 4$.
3. States $DE = 16$.
4. States $EB = 8$.
5. Computes $AE = AD + DE = 20$.
6. Computes $DB = DE + EB = 24$.
7. Computes $AB = AD + DE + EB = 28$.
8. States the ordering/collinearity $A - F - G - C$ on $\overline{AC}$ (global setup).
9. States $AF = 13$.
10. States $FG = 52$.
11. States $GC = 26$.
12. Computes $AG = AF + FG = 65$.
13. Computes $FC = FG + GC = 78$.
14. Computes $AC = AF + FG + GC = 91$.
15. Defines $M$ as the reflection of $D$ across $F$ (so $F$ is midpoint of $DM$).
16. Elaborates reflection at $F$; observes $F$ midpoint of $DM \Rightarrow \text{Area}(AFM) = \text{Area}(ADF)$.
17. Defines $N$ as the reflection of $G$ across $E$ (so $E$ is midpoint of $GN$).
18. Notes explicitly that $E$ is the midpoint of $GN$.
19. States $\text{Area}(DEGF) = 288$.
20. Sets $\text{Area}(ADF) = \frac{AD}{AB} \cdot \frac{AF}{AC} \cdot \text{Area}(ABC) = \frac{S}{49}$.
21. Sets $\text{Area}(AEG) = \frac{AE}{AB} \cdot \frac{AG}{AC} \cdot S = \frac{25S}{49}$.

22. Relates $\text{Area}(DEGF) = \text{Area}(AEG) - \text{Area}(ADF) = \frac{24S}{49}$.

23. Solves $\frac{24S}{49} = 288 \Rightarrow S = \text{Area}(ABC) = 588$.

24. Computes $\text{Area}(ADF) = S/49 = 12$.

25. Uses reflection at $F \Rightarrow \text{Area}(AFM) = \text{Area}(ADF)$.

26. Concludes $\text{Area}(AFM) = 12$.

27. Computes $\text{Area}(ABG) = \frac{AG}{AC} \cdot S = 420$.

28. Computes $\text{Area}(BEG) = \text{Area}(ABG) - \text{Area}(AEG) = 420 - 300 = 120$.

29. Uses reflection at $E \Rightarrow \text{Area}(BEN) = \text{Area}(BEG)$.

30. Concludes $\text{Area}(BEN) = 120$.

31. Chooses coordinates $A = (0,0), B = (28,0), C = (0,42)$; derives $D, E, F, G, M, N$ coordinates.

32. Applies the shoelace formula to the heptagon $AFNBCEM$ and gets area 588.

33. States the final result: $\boxed{588}$.

We can observe that the DAG has 8 non-closed nodes. We diagnose the reasons of uncloseness for each node:

- **Nodes** 1 **&** 8: These two nodes state the global setup on collinearity/ordering, which are directly provided by the problem statement. The later steps implicitly use collinearity to add segment lengths on $\overline{AB}$ and $\overline{AC}$, but the LLM does not recognize that it has used these two nodes in its subsequent reasoning.

- **Nodes** 6 **&** 13 **&** 16 **&** 18: These nodes derive or state extra commentary of their previous step, which are not needed in the subsequent steps.

Then, the message conveyed by **Nodes** 26 **&** 30 is *crucial*. In this problem, the area of the heptagon $AFNBCEM$ can be computed via two distinct strategies:

- **Reflection-swap strategy**: The core idea is to replace two interior triangles ($\triangle ADF$, $\triangle BEG$) of $\triangle ABC$ with their exterior reflected counterparts ($\triangle AFM$, $\triangle BEN$), showing that the net area change is zero. Consequently, the heptagon's area is obtained by a straightforward "remove + add" bookkeeping.

- **Shoelace strategy**: This coordinate-based, algebraic method requires only listing the vertices $A, F, N, B, C, E, M$ in order and then applying the determinant sums to compute the area.

Nodes 20 through 30 derive the areas required for the final "remove + add" computation in the reflection-swap strategy, namely

$$\text{Area}(AFNBCEM) = \underbrace{\text{Area}(ABC)}_{\textbf{Node } 23} - \underbrace{\text{Area}(ADF)}_{\textbf{Node } 24} - \underbrace{\text{Area}(BEG)}_{\textbf{Node } 28}$$
$$+ \underbrace{\text{Area}(AFM)}_{\textbf{Node } 26} + \underbrace{\text{Area}(BEN)}_{\textbf{Node } 30} .$$

If the model had continued with this strategy, **Node 31** would correspond to the above equation, with **Nodes** 26 **&** 30 as its parents. However, the model instead switches to the shoelace strategy at **Node 31** and successfully obtains the correct answer, leaving **Nodes** 26 **&** 30 unclosed.

This provides evidence that the model generates elements of an alternative strategy that remain unused in the current trajectory and switches strategies during the generation process.

Next, the second and third trajectories exhibit node structures similar to the first. They also contain nodes such as **Nodes** 1 **&** 8 in the first trajectory, which are not recognized as being used. However, unlike the first trajectory, they rely solely on the reflection-swap strategy to obtain the final answer without switching strategy.

The final trajectory consists of 121 nodes in total. We provide a comprehensive review of its reasoning process: it begins by copying the givens, constructing segment sums, and recording the reflection, analogous to **Nodes** 1–19 in the first trajectory. It then attempts a parametric area strategy via trigonometric parametrization but halts after approximately 20 steps. Subsequently, it searches over many polygon decompositions of heptagon $AFNBCEM$, repeatedly proposing and discarding formulas—clear evidence of exploratory search—until it identifies the structural invariant $AD : DE : EB = AF : FG : GC = 1 : 4 : 2$, from which it correctly infers $DF \parallel EG \parallel BC$. At this stage, the strategy shifts to a similarity/area-ratio approach for triangles sharing angle $A$ and successfully derives the area of $\triangle ABC$. Finally, the strategy switches once more to the reflection-swap method, yielding the correct answer.

## B.2 AN IMO-LEVEL EXAMPLE DAG

There are several ways in which an IMO-level proof can fail the logical-closeness test even when substantial portions of the argument are correct:

- a model may solve the fixed-point subproblem and mistake fixed points for all valid starting values;
- a model may prove the forward dynamics for $N \in \mathcal{S}$ but omit the complementary necessity argument for $N \notin \mathcal{S}$;
- a model may derive both macro-branches but never fuse them into a single final characterization.

For such problems, a lemma-level DAG is more informative than a step-level DAG. We illustrate this with the following problem from IMO 2025.

---

**IMO 2025 Q4 Problem**

A proper divisor of a positive integer $N$ is a positive divisor of $N$ other than $N$ itself.

The infinite sequence $a_1, a_2, \ldots$ consists of positive integers, each of which has at least three proper divisors. For each $n \geq 1$, the integer $a_{n+1}$ is the sum of the three largest proper divisors of $a_n$.

Determine all possible values of $a_1$.

---

The rigorous solution shows that the valid starting values are exactly

$$\mathcal{S} = \{N : v_2(N) \text{ is odd}, \ v_3(N) \geq \tfrac{v_2(N)+1}{2}, \ 5 \nmid N\}.$$

The resulting lemma-level task-specific DAG drawn from `Claude-Opus-4-6` is shown in Fig. 7.

We summarize each node's conclusion below.

**1** Defines the dynamical map $f(N)$, validity of a starting value, and the candidate set $\mathcal{S}$.

**2** Proves the divisor-pairing identity $f(N) = N\left(\frac{1}{s_2} + \frac{1}{s_3} + \frac{1}{s_4}\right)$.

**3** Solves $\frac{1}{a} + \frac{1}{b} + \frac{1}{c} = 1$ in strictly increasing integers and obtains the unique triple $(2, 3, 6)$.

**4** Characterizes the fixed points: $f(N) = N$ if and only if $N = 2 \cdot 3^b \cdot m$ with $b \geq 1$ and $\gcd(m, 30) = 1$.

**5** Shows that odd inputs stay odd and strictly decrease under $f$.

**6** Shows that when $12 \mid N$ and $5 \nmid N$, one has $f(N) = \frac{13N}{12}$ and $(v_2, v_3) \mapsto (v_2 - 2, v_3 - 1)$.

**7** Proves sufficiency: every $N \in \mathcal{S}$ is valid.

**8** Proves forward invariance of the complement: $N \notin \mathcal{S} \Rightarrow f(N) \notin \mathcal{S}$.

**9** Proves necessity: every $N \notin \mathcal{S}$ eventually becomes invalid.

**L** Correct sink: the valid starting values are exactly the elements of $\mathcal{S}$.

**L′** Incorrect sink: the valid starting values are exactly the fixed points of $f$.

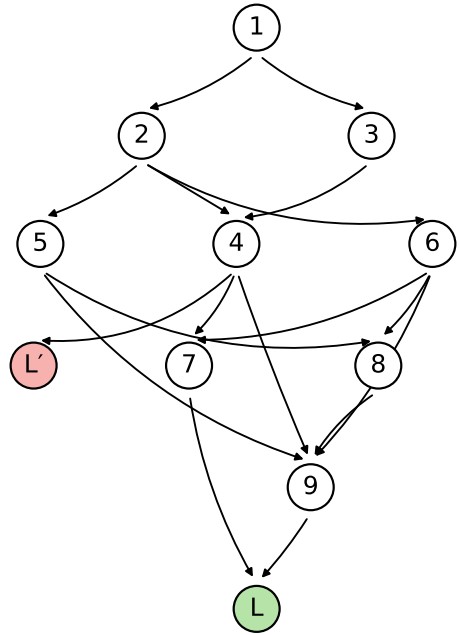

Figure 7: Lemma-level task-specific DAG for IMO 2025 Q4. The green sink is the correct theorem, while the red sink is a plausible but incorrect premature termination.

The green sink requires both macro-branches of the proof. On the left, Nodes 2–4–6–7 show that numbers in $\mathcal{S}$ are not necessarily fixed at time 0, but repeated use of Node 6 decreases $v_2$ by 2 and $v_3$ by 1 until the orbit reaches the fixed-point family from Node 4. Concretely, if

$$N = 2^{2k+1} \cdot 3^b \cdot m \in \mathcal{S} \qquad (b \geq k+1,\ \gcd(m,30) = 1),$$

then after $k$ applications one obtains

$$f^k(N) = \left(\frac{13}{12}\right)^k N = 2 \cdot 3^{b-k} \cdot 13^k m,$$

which lies in the fixed-point family and therefore generates an infinite valid orbit.

On the right, Nodes 5, 6, 8, and 9 show that the complement of $\mathcal{S}$ cannot support an infinite orbit. Node 8 keeps every iterate outside $\mathcal{S}$. Then Node 9 selects the minimum element of the orbit and proves that any local growth pattern must fall into either the short-lived $(2, 3, 5)$ regime or the $(2, 3, 4)$ regime; both eventually force strict descent, contradicting well-foundedness of $\mathbb{N}$.

The incorrect sink $L'$ models a natural but wrong termination. After Nodes 2, 3, and 4, a model might conclude that the only infinite sequences are the fixed points of $f$. This is a coherent subproof, but it misses the genuinely valid *preperiodic* starting values in $\mathcal{S}$ with $v_2(N) \geq 3$, which are not fixed at time 0 and only flow to a fixed point after finitely many applications of Node 6.

This failure mode is especially useful for evaluating logical closeness. A trajectory ending at $L'$ has solved a substantial subproblem-the complete classification of fixed points-yet it omits the lemmas that distinguish *fixed points* from *all valid starting values*. In other words, $L'$ is not a random wrong answer; it is a mathematically polished but globally incomplete termination.

At the DAG level, the proof therefore separates into a left sufficiency branch and a right necessity branch, both of which are indispensable for the correct sink. A logically closed trajectory must collect enough ancestors to justify Node L, rather than stopping as soon as Node 4 is established. For IMO-level arguments of this type, the lemma-level DAG exposes the key structural distinction much more clearly than a long list of atomic algebraic steps.

## C   EXAMPLE DAG-MATH FORMATTED CoT

> **Example DAG-MATH Formatted CoT for the logarithmic count problem**
>
> **Step 1**
>
> **Edge**: Restate the target equation from the problem statement, $2\log(x-1) = \log k$, which is the central equation to solve for $x$ in terms of $k$.
> **Parents**: Null
> **Node**: $2\log(x-1) = \log k$.
>
> $\vdots$
>
> **Step 4**
>
> **Edge**: Since the left-hand side of the equation in Step 1 contains $\log(x-1)$, the domain restriction for a logarithm requires $x - 1 > 0$, i.e., $x > 1$.
> **Parents**: Step 1
> **Node**: $\log(x-1)$ requires $x > 1$.
>
> $\vdots$
>
> **Step 8**
>
> **Edge**: Using Step 7, the candidate solutions are $1 \pm \sqrt{k}$; Step 4 requires $x > 1$, and Step 5 ensures $k > 0$, so $\sqrt{k} > 0$, making $1 - \sqrt{k} < 1$ invalid. Therefore, $1 + \sqrt{k}$ is the only admissible solution.
> **Parents**: Step 4,5,7
> **Node**: $1 + \sqrt{k}$ is the only admissible solution.
>
> **Step 9**
>
> **Edge**: From Step 8, each positive integer $k$ yields exactly one solution $x = 1 + \sqrt{k}$. Step 5 requires $k > 0$, and Step 2 restricts $k$ to integers in $[1, 300]$. Therefore, there are 300 valid $k$ values, satisfying Step 3's requirement of exactly one solution.
> **Parents**: Step 2,3,5,8
> **Sink Node**: There are $\boxed{300}$ valid values for $k$.

## D   BENCHMARK CONSTRUCTION AND VALIDATION

In this section, we detail our three-stage strategy for benchmark construction with stage-wise validation method.

**Stage 1:** We prompt `GPT-o4-mini` to generate *only* the Node set, one step at a time, using the instructions in Appendix D.1. To enhance correctness, we adopt a supervised setup that provides both the problem and its correct solution. Because standard CoT can skip arithmetic or combine multiple results in a single step, we require that each Node consist of exactly one sentence containing a single mathematical or logical assertion (i.e., one primitive action per step) to normalize granularity.

**Validation 1** The intermediate stepwise correctness and final answer correctness are validated using both SymPy and LLM-as-Judge (via `GPT-4o`; if the final answer or any intermediate assertion is incorrect, the complete Node set is rejected and resampled. We iteratively repeat this procedure until passing all checks.

**Stage 2:** Given the verified Node set from Stage 1, we prompt `GPT-o4-mini` (per Appendix D.2) to assign, for each Node, a minimal set of direct Parents sufficient to derive it via a primitive operation. We enforce acyclicity and well-typed arity constraints, then assemble the full DAG.

**Validation 2** The resulting DAG is checked for logical closeness relative to the sink node; if checks fail, all dependencies are resampled. We repeat this procedure for 5 times. If the DAG is still not closed, then we go back to Stage 1 to regenerate Node

set. If the Node set of second turn still fails to close for 5 times, we filter out the problem. Next, we employ `GPT-o3`[5] to evaluate the correctness dependency structure.

**Stage 3:** Conditioned on the generated [Parent(s), Node] pairs, we prompt `Qwen3-235B`[6] (per Appendix D.3) to generate the Edge content that justifies how each Node is inferred from its Parents. The justification must introduce no new facts beyond the problem statement and the cited Parents. After this step, the triplets are merged into a gold-standard DAG-MATH formatted CoT in forward order.

Table 2 reports the success rate over problem difficulty after three-stage construction with validation on Omni-MATH. We limit the problem difficulty within 6 to ensure the reliability of benchmark (IMO-level problem beyond 6).

Table 2: Benchmark Construction Success Rate by Problem Difficulty

| Difficulty | Total | Success | Rate |
|---|---|---|---|
| (0, 1] | 124 | 119 | 96.0% |
| (1, 2] | 363 | 343 | 94.5% |
| (2, 3] | 197 | 182 | 92.4% |
| (3, 4] | 746 | 665 | 89.1% |
| (4, 5] | 1304 | 1072 | 82.2% |
| (5, 6] | 649 | 498 | 76.7% |

## D.1 PROMPT FOR STAGE 1

---
**Stage 1 - Instructions for Refined Step-by-Step Answer**

**Task:**

You are an expert in mathematical logic and reasoning. Your job is to take rough multi-step math solutions and rewrite them in a detailed, structured, and logical step-by-step format. Follow these guidelines:

- Each step must be exactly one sentence, and that sentence may contain only one mathematical or logical assertion.
- Do not combine multiple assertions in one step.
- Show every tiny inference—setting up equations, solving for variables, converting units, etc.—each in its own step.
- Any fact, formula, problem detail, or intermediate result must itself be stated in its own individual step.
- Avoid including irrelevant steps that do not contribute to the solution.
- The final step should be the answer statement in the form: `The final answer is \boxed{xxx}`.
- Use LaTeX format enclosed in dollar signs for all mathematical expressions.

**Problem:** `{problem statement}`

**Solution:** `{original step-by-step solution}`

**Refined Step-by-Step Answer:**

---

## D.2 PROMPT FOR STAGE 2

---
**Stage 2 - Instructions for Step Dependency Analysis**

**Role and Objective**
You are an expert in mathematical logic and stepwise reasoning. Your primary role is to analyze the detailed solution steps for a mathematical problem and annotate each step with its minimal set of direct dependencies.
**Instructions**

---

[5] `GPT-o3` is a stronger model than `GPT-o4-mini`
[6] We choose `Qwen3` as it provides a clearer description of the inference from Parents to Nodes than `GPT-o4`.

1. **Input Structure:**
   – Each problem appears as a JSON object with fields:
   `problem_text` (string): Problem description.
   `final_answer` (string): The solution.
   `steps` (array): Each is an object containing:
     `step_id` (integer): Unique step identifier.
     `text` (string): Reasoning or mathematical operation.

2. **Dependency Annotation:**
   – Add a `direct_dependent_steps` field for every step.
   – For each step:
     If stated directly from the problem statement, assign `null`.
     Otherwise, list the minimal, directly required prior `step_id` values in *ascending order*, e.g., `[2, 3]`.
   – **Dependency Rules:**
     1. Every dependency ID in `direct_dependent_steps` **MUST** exist in the original set of `step_id` values of the input.
     2. Every listed dependency must be a prior step—its `step_id` must be strictly less than the current step (i.e., no self- or future-dependency is allowed).

3. **Self-Validation for Step Closure:**
   – After assigning dependencies, check for unclosed intermediate steps:
     – An unclosed step is any non-final step not referenced in `direct_dependent_steps` by any subsequent step.
     – If any exist, refine dependencies until all intermediate steps are "closed" (each is used at least once by a later step).

4. **Post-action Validation:**
   – After annotating dependencies and closing all steps, validate that each intermediate step is referenced at least once by a subsequent step before finalizing the output. If any issues are found, self-correct and repeat the closure process.

5. **Structured Output and Consistency:**
   – The number of steps in the structured output **MUST** match exactly the number of steps in the original input.

**Application Process**
– When presented with a new problem in valid JSON:
   1. Iterate through steps in order of `step_id`.
   2. For each step, determine if it relies on previous steps or the problem statement and annotate `direct_dependent_steps` as specified.
   3. Validate dependencies (all dependency IDs exist; all point to a prior step).
   4. Check for unclosed steps and adjust as necessary.
   5. Validate closure and present the final modified problem.

**Error Handling**
– If the input is malformed (e.g., required fields missing, `step_id` missing, or `step_id` values not strictly ascending):
   – Return only a JSON object with an `error` field and a concise message. For example: `"error": "Input data malformed: missing step_id in step 3."`

**Output Format**
– Output the structured response with all steps mirroring the original order and count, each annotated with `direct_dependent_steps`. If there is an error, output only the error object as described.
```
{JSON text of refined step-by-step answer with problem
statement from Stage 1}
```

## D.3 PROMPT FOR STAGE 3

---

**Stage 3 - Instructions for Constructing Edge Inference**

**Role and Objective**

- You serve as a mathematics solution explainer. For each step in a solved mathematics problem (provided as structured JSON), generate a detailed and explanatory justification paragraph (`edge` string) clarifying the correctness and logical progression.

**Instructions**

- For each step in the provided `"steps"` array within the problem JSON, write a `edge` string that satisfies the rules below.

- If `direct_dependent_steps` is not `null`, explicitly justify how `direct_dependent_steps` support the current one, citing their IDs. If `null`, note that the step is given by the problem statement or background knowledge (definition, theorem, lemma, fact, or general knowledge not originally in the problem statement).

- Clearly explain the mathematical or logical principle, operation, or procedure used (e.g., counting rule, algebraic manipulation, definition, theorem, arithmetic operation) applied in the step.

- Strict Rule: Do not omit any referenced dependency, the `edge` **MUST** include every step in `direct_dependent_steps`.

- Clearly illustrate why we need to do this step and how we arrive at this step (acts like planning).

- For numeric calculations, perform the arithmetic clearly and include a brief sanity check as appropriate.

- Use neutral, present-tense, explanatory sentences with active voice.

- Ensure each justification is self-contained, needing only the current step and referenced prior steps to be understandable.

**Context**

- Input: JSON object representing a solved math problem with an array of steps, each with `step_id`, `text`, and `direct_dependent_steps`.

- Output: Return only the `edge` for each step, ordered to match the original steps array.

- Do **NOT** include internal model reasoning or any execution commentary in the output.

**Demonstration Example**

Here is a demonstration of the style and format of the edge field. You need to follow this style and format.

- "We plan to exclude numbers divisible by 2, 3, or 5. To do that systematically we first express the count of multiples of each relevant divisor in the domain. The number of multiples of $k$ up to $n$ is $\left\lfloor \frac{n}{k} \right\rfloor$. Applying that with $n = 999$ (from Step 1) and $k = 2$ gives $\lfloor 999/2 \rfloor$. Writing it as a floor expression is precise and handles the non-divisible endpoint gracefully."

- "Since numbers divisible by 2, 3, and 5 are overlapping, we need to subtract the count of numbers divisible by pairwise combinations of 2, 3, and 5 then add back the count of numbers divisible by all three, i.e., inclusion-exclusion. The least common multiple of 2 and 3 is 6, so count multiples of 6. Recall we have 999 integers in the domain from Step 1, count multiples of 6 up to 999 via $\lfloor 999/6 \rfloor$. This uses the same floor-division approach but applied to the least common multiple of the pair."

- "We convert the expression in Step 8 into a concrete number to use in later arithmetic. Compute $999/6 = 166.5$; taking the floor yields 166. Quick cross-check: $166 \times 6 = 996$, so the last multiple is exactly 996."

- "We now have all the building blocks: counts of singles, pairwise intersections, and the triple intersection. The immediate plan is to apply the inclusion-exclusion formula for

---

three sets to compute the size of the union $A \cup B \cup C$ where $A =$ multiples of 2, $B =$ multiples of 3, $C =$ multiples of 5. Inclusion-exclusion avoids overcounting and is the rigorous combinatorial tool for unions of overlapping sets. The three-set inclusion-exclusion identity is $|A \cup B \cup C| = |A| + |B| + |C| - (|A \cap B| + |A \cap C| + |B \cap C|) + |A \cap B \cap C|$. Substitute the numerical values computed from Steps 3, 5, 7, 9, 11, 13, 15,: $499 + 333 + 199 - (166 + 99 + 66) + 33$. Writing it explicitly as $499 + 333 + 199 - 166 - 99 - 66 + 33$ lays out the arithmetic to be performed next."

- "Cite the standard definition: 1 has exactly one positive divisor (1 itself), so it is neither prime (requires two distinct divisors) nor composite (requires more than two). In partitioning the 266 numbers not divisible by 2,3,5 into primes and composites, we must also handle the special case 1, which is counted among the 266 but is neither prime nor composite; failing to account for 1 would misclassify one element."

- "We now simplify the complex expression from Step 12, which is $(48 \times 15) - (320 \div 8) + 27$. The order of operations dictates multiplication and division before addition or subtraction. First compute $48 \times 15 = 720$. Next compute $320 \div 8 = 40$. Substituting these back into the expression gives $720 - 40 + 27$. This reduction preserves equivalence while making the next step—performing the remaining subtraction and addition—more straightforward."

**Core Language Style Requirements**

- Match the tone, sentence flow, and level of detail in the demonstration examples provided below.

- Write in a way that reads naturally to a human reasoner, not like a formal audit log.

**Self-Validation**

- **IMPORTANT**: After each `edge` is generated, validate that whether it includes every step in `direct_dependent_steps`. If not, you need to re-generate the `edge` for the step. Notice that you need to check for all steps, not just a subset of steps.

- For each step, internally determine the applicable mathematical principle, identify dependencies, clarify evaluation or logic, and succinctly state both justification and role in the broader problem.

- Clearly state mathematical justifications, and follow the required language style. If a numeric evaluation is performed, confirm a brief sanity check is present.

**Verbosity**

- `edge` should be **detailed**, allowing for expansion if the step is complex or multi-part.

**Stop Conditions**

- Complete when every step has a well-justified `edge` field, following all style and content rules. Escalate for ambiguous or incomplete input only.

**Output Format**

- Return only a `edge` field for each step populated as described. Do not include extra fields, wrappers, or commentary.

EXAMPLE JSON OUTPUT:

```
{
    "edges": [
      {
        "step_id": 1,
        "edge": "We define ..."
      },
      {
        "step_id": 2,
        "edge": "Building on the definition of ..."
      },
      {
        "step_id": 3,
```

```
        "edge": "Using the expression for ..."
    }
}
{JSON text of refined step-by-step answer with problem
statement from Stage 1 and dependencies from Stage 2}
```

## E  HUMAN EVALUATION OF GOLD-STANDARD BENCHMARK

To validate the reliability of the DAG-MATH benchmark and address potential concerns regarding the subjectivity of human or LLM-annotated dependencies, we designed an algorithmic human evaluation pipeline. This approach treats the logical structure of a mathematical solution as isomorphic to a computational graph, where "dependencies" correspond to the flow of variables in an executable program.

**Pipeline** We formalize the validation process through the following four-step pipeline, applied to a random sample of 50 problems from the benchmark:

1. check basic stepwise correctness and final answer correctness **by human**,
2. map each natural language step (node) to a functional line(s) of SymPy code **by human**,
3. execute the complete code to test validity then extract the computational graph of code. If the code for step_k referenced the variable step_j, we record a computational dependency.
4. compare the set of computational dependencies extracted from the code against the direct_dependent_steps.

For steps which cannot be converted to SymPy code such as declarative constraint (e.g., "an arrangement of hats corresponds to a permutation of the three people"), or strategy (e.g., "we propose to count the total arrangements and subtract the invalid ones"), we validate the dependencies by human knowledge. We provide a complete human-evaluated example for illustration in Appendix E.1.

**Results** All samples pass the correctness check in Step 1 and 49/50 samples pass our dependency check. The sample that not pass contains one irrelevant node, i.e. "Lauren plays basketball with her friends", which is closed but has very weak connection to its child ("Lauren makes 10 baskets"). Our human evaluation confirms the benchmark's high reliability, achieving 100% execution correctness and 98% dependency accuracy, with the sole outlier attributable to a weak semantic link in a narrative context step rather than a structural reasoning failure.

### E.1  HUMAN EVALUATION EXAMPLE

> **Rato of Distance Problem**
>
> Natascha cycles 3 times as fast as she runs. She spends 4 hours cycling and 1 hour running. What is the ratio of the distance that she cycles to the distance that she runs?

```python
from sympy import symbols, simplify, Rational

def solve_problem_42():

    # Step 1: "Natascha cycles 3 times as fast as she runs."
    # Dependencies: [null]
    step_1 = 3

    # Step 2: "Natascha spends 4 hours cycling."
    # Dependencies: [null]
    step_2 = 4

    # Step 3: "Natascha spends 1 hour running."
```

```
14        # Dependencies: [null]
15        step_3 = 1
16
17        # Step 4: "Let r represent Natascha's running speed in km/h."
18        # Dependencies: [null]
19        # We initialize the symbolic variable.
20        step_4 = symbols('r')
21
22        # Step 5: "Cycling speed is 3r km/h."
23        # Dependencies: [1, 4]
24        # Multiplier (step_1) * Running Speed (step_4)
25        step_5 = step_1 * step_4
26
27        # Step 6: "The formula for distance... is d=vt."
28        # Dependencies: [null]
29        # We define the distance function here.
30        step_6 = lambda v, t: v * t
31
32        # Step 7: "Running distance... yields d = r * 1."
33        # Dependencies: [3, 4, 6]
34        # Apply Formula (step_6) to Speed (step_4) and Time (step_3)
35        step_7 = step_6(v=step_4, t=step_3)
36
37        # Step 8: "Simplifying... running distance as r."
38        # Dependencies: [7]
39        # Simplify previous result
40        step_8 = simplify(step_7)
41
42        # Step 9: "Cycling distance... yields d = 3r * 4."
43        # Dependencies: [2, 5, 6]
44        # Apply Formula (step_6) to Speed (step_5) and Time (step_2)
45        step_9 = step_6(v=step_5, t=step_2)
46
47        # Step 10: "Simplifying... cycling distance as 12r."
48        # Dependencies: [9]
49        # Simplify previous result
50        step_10 = simplify(step_9)
51
52        # Step 11: "Ratio... is 12r : r."
53        # Dependencies: [8, 10]
54        # Construct the fraction (Ratio) using Running (step_8) and Cycling (
              step_10)
55        step_11 = (step_10, step_8)
56
57        # Step 12: "Simplifying... gives 12:1."
58        # Dependencies: [11]
59        # Divide the terms in the ratio
60        ratio_val = step_11[0] / step_11[1]
61        step_12 = simplify(ratio_val)
62
63        # Step 13: "The final answer is \boxed{12:1}."
64        # Dependencies: [12]
65        # Format the result from step_12
66        step_13 = f"The final answer is \\boxed{{{step_12}:1}}."
67
68        return step_13
```

## F  FEW-SHOT PROMPT FOR DAG-MATH FORMATTED CoT

---

**Instructions for DAG-MATH Formatted CoT Generation**

**Role and Objective**

You are an expert mathematical reasoner and logician. For the given problem, you must produce a single, valid JSON object containing a list of solution steps. Each step in the list must be an object with the following four fields: `step_id`, `edge`, `direct_dependent_steps`, and `node`.

**Requirements**

Each step has **exactly**:

- `step_id` (int; unique; strictly increasing)

- `edge` (sentences; describe *why/how* this step follows; reference prior steps with Step's `step_id` tags, e.g., `Step 1`, `Step 3`)
    - State the goal of the current step.
    - Cite the *minimal* **direct** dependent previous steps used, with *how* these steps are used for the current step. **IMPORTANT**: every direct dependent step *must* be cited in the form Step's `step_id`.
    - State clearly the mathematical principle being applied (e.g., inclusion–exclusion, algebraic manipulation, definition), which turns those inputs into the asserted output.

- `direct_dependent_steps` (array of ints **or** `null`)
    - This field must contain a list of `step_id`s representing the minimal set of prior steps directly used to derive the current step in `edge`.
    - The list must be in ascending order (e.g., `[2, 5]`).
    - If a step is a fact taken directly from the problem statement, this field should be `null`.
    - Topological order: every dependency ID < current `step_id`.
    - Closure: every nonfinal step's `step_id` must appear in some later step's `direct_dependent_steps`.

- `node` (one execution sentence)
    - Each step's `node` field must contain a **single, atomic** sentence making exactly one logical assertion (e.g., stating an equation, defining a variable, presenting a calculation result) which acts as the results inferred from `edge`.
    - All information, including facts from the problem statement and intermediate results, must be broken down into these atomic steps.
    - Avoid including irrelevant steps that do not contribute to the solution.
    - The final step **must** be the answer statement in the form: "The final answer is $\boxed{\cdots}$".

**Global Constraints**

- Use LaTeX format enclosed in dollar signs for all mathematical expressions.

- Your entire output must be a single, valid JSON object. Do not include any text or commentary outside of the JSON structure. You will be provided with high-quality examples to demonstrate the required format and reasoning style.

**Demonstration Examples**

`{Gold-standard demonstration examples}`

**Bad Examples**

1. The example below is bad since Step's `step_id` are missing in `edge`.

```
{
  "steps": [
    ...
    {
```

---

```
          "step_id": 36,
          "edge": "Using the confirmed digit values...",
          "direct_dependent_steps": [
            8, 15, 26, 32, 35
          ],
          "node": "The sum of the digits $K + L + M + N$
            is $0 + 5 + 3 + 9$."
        },
        ...
      ]
    }
```

2. The example below is bad since plural "Steps 8, 9, and 10" is used in `edge` instead of singular "Step 8, Step 9, and Step 10".

```
{
  "steps": [
    ...
    {
      "step_id": 11,
      "edge": "From Steps 8, 9, and 10, we have...",
      "direct_dependent_steps": [
        8, 9, 10
      ],
      "node": "Define d = b - a and e = c - a. Then
        the equations become: (a + e) * d = 1 + 2k,
        (a + d) * e = -3 + 6k, a * (e - d) = -4 + 4k."
    },
    ...
  ]
}
```

**Problem:** {Test problem statement}
**Solution:**

# G    PROMPTS FOR SENSITIVITY ANALYSIS

## G.1    PROMPT WITH DIFFERENT FORMAT

**Instructions for DAG-MATH Solution Generation**

**Role and Objective:** You are an expert mathematical reasoner and logician. For any given problem, you must produce a single, valid JSON object containing a list of solution steps.

**Requirements:**
Output one JSON object: { `"steps": [ ...]` }. Each step specification:

| Field Name | Data Type | Requirements | Description |
|---|---|---|---|
| step_id | Integer | • Must be unique
• Must be strictly increasing | Unique identifier for each step in ascending order |
| thinking | String | • Describe *why/how* this step follows
• Reference prior steps as **Step step_id** (e.g., Step 1, Step 3)
• Must include:
  – Goal of current step
  – Citation of minimal **direct** dependent steps with *how* they're used
  – Mathematical principle being applied | Reasoning that explains the logical derivation of this step |
| direct_dependent_steps | List OR null | • List of step_id values
• Must be in ascending order (e.g., [2, 5])
• Use null if fact from problem statement
• Every dependency ID < current step_id
• Every non-final step must appear in some later step's list | Minimal set of immediate prior steps required for current step |
| text | String | • Single, atomic sentence
• Makes exactly one logical assertion
• Acts as result inferred from thinking
• Final step format: "The final answer is ⬚..." | Execution statement presenting the conclusion of this step |

**Closure Constraint:** Every non-final step's step_id must appear in at least one subsequent step's direct_dependent_steps array.

**Global Constraints:**

- Use LaTeX format enclosed in dollar signs for all mathematical expressions
- Output must be a single, valid JSON object only
- No text or commentary outside the JSON structure
- High-quality examples will be provided to demonstrate the required format and reasoning style

**Demonstration Examples**
{Gold-standard demonstration examples}

**Bad Examples**
{Bad demonstration examples}

## G.2 Prompt with Rephrasing

---

**Instructions for DAG-MATH Solution Generation**

**Role and Objective:** Your function is that of a specialist in mathematical reasoning and logical analysis. When presented with any problem, your task is to generate a single, properly-formed JSON object that contains an array of solution steps. Within this array, every step must be represented as an object containing these four specific fields: `step_id`, `thinking`, `direct_dependent_steps`, and `text`.

**Requirements:** Generate a JSON object with this structure: `{ "steps": [ ...] }`. The specifications for each step are as follows:

- `step_id` (integer value; no duplicates; monotonically increasing sequence)

- `thinking` (prose description; articulate the *reasoning* and *methodology* behind this step; make references to previous steps using **Step step_id** notation, such as `Step 1` or `Step 3`)
    - Articulate the objective that this particular step accomplishes.
    - Identify the *smallest set* of **immediately preceding** steps that are utilized, explaining the *manner* in which these steps contribute to the current one. **CRITICAL**: each immediately preceding step *must* be mentioned using the `Step step_id` format.
    - Explicitly state the mathematical concept or operation being employed (e.g., the principle of inclusion-exclusion, manipulation of algebraic expressions, application of a definition), which transforms the referenced inputs into the claimed result.

- `direct_dependent_steps` (list of integers **or** `null` value)
    - This field is required to hold an array of `step_id` values that represent the smallest collection of prior steps that must be available to derive the present step.
    - The array must maintain ascending numerical order (for instance, `[2, 5]`).
    - When a step represents information extracted directly from the problem description, this field should contain `null`.
    - Topological ordering requirement: each dependency identifier $<$ the current `step_id`.
    - Closure property: each non-terminal step's `step_id` is required to be listed within the `direct_dependent_steps` of at least one subsequent step.

- `text` (single sentence for execution)
    - The `text` field of each step is required to contain a **single, indivisible** sentence that makes precisely one logical claim (such as declaring an equation, introducing a variable definition, or stating a computational outcome) which serves as the conclusion derived from `thinking`.
    - Every piece of information, encompassing facts extracted from the problem description and intermediate computational results, is required to be decomposed into these indivisible steps.
    - Exclude any irrelevant steps that fail to advance the solution.
    - The terminal step **is required to** take the form of an answer declaration: "The final answer is $\boxed{...}$".

**Global constraints:** Employ LaTeX notation surrounded by dollar signs for every mathematical expression.

The entirety of your output is required to be a single, well-formed JSON object. Exclude any textual content or explanatory remarks beyond the JSON structure. High-quality exemplars will be supplied to illustrate the expected format and reasoning approach.

**Demonstration Examples**
`{Gold-standard demonstration examples}`

---

> **Bad Examples**
> `{Bad demonstration examples}`

## H   PROMPTS FOR DAG PARSING

### H.1   PROMPT FOR THINKING LLMs

> **Instructions for Step Dependency Analysis**
>
> **Role and Objective**
> You are an expert in mathematical logic and stepwise reasoning. Your primary role is to analyze the detailed solution steps for a mathematical problem and annotate each step with its minimal set of direct dependencies.
> Begin with a concise planning checklist (3-7 bullets) of what you will do; keep items conceptual, not implementation-level.
> **Instructions**
>
> 1. **Input Structure:** Each problem appears as a JSON object with fields:
>    - `problem_text` (string): Problem description.
>    - `steps` (array): Each is an object containing:
>      - `step_id` (integer): Unique step identifier.
>      - `text` (string): Mathematical operation/action of this step.
>
> 2. **Dependency Annotation:**
>    - Add `direct_dependent_steps` for every step.
>    - For each step:
>      - If derived only from the problem statement, assign 'null'.
>      - Otherwise, list the minimal, immediately required prior `step_id` values in ascending order, e.g., '[2, 3]'.
>
> 3. **Dependency Rules:**
>    - Every dependency ID in `direct_dependent_steps` MUST exist in the original set of `step_id` values of the input.
>    - Every listed dependency must be a prior step—its `step_id` must be strictly less than the current step (i.e., no self- or future-dependency is allowed).
>
> 4. **Structured Output and Consistency:**
>    - The number of steps in the structured output MUST match exactly the number of steps in the original input.
>    - The output must strictly follow the schema `DependencyGraph`.
>
> **Error Handling**
>
> 1. If the input is malformed (e.g., required fields missing, `step_id` missing, or `step_id` values not strictly ascending):
>
> 2. Return only a JSON object with an `error` field and a concise message. For example:
>
>    ```
>    {"error": "Input data malformed:
>    missing step_id in step 3."}
>    ```
>
> **Output Format**
> Output the structured response with all steps mirroring the original order and count, each annotated with `direct_dependent_steps`. If there is an error, output only the error object as described.

## H.2 Prompt for Non-Thinking LLMs

---

**Instructions for Step Dependency Analysis**

**Role and Objective**

You are an expert in mathematical logic and stepwise reasoning. Provide a reason for each step to explain its minimal set of direct dependencies then annotate each step with its dependencies.

Begin with a concise planning checklist (3-7 bullets) of what you will do; keep items conceptual, not implementation-level.

**Instructions**

1. **Input Structure:** Each problem appears as a JSON object with fields:
   - `problem_text` (string): Problem description.
   - `steps` (array): Each is an object containing:
     - `step_id` (integer): Unique step identifier.
     - `text` (string): Mathematical operation/action of this step.

2. **Dependency Annotation:** For each step, produce:
   - `reason`: An explanation justifying the direct dependencies, referencing step IDs or the problem statement.
   - `direct_dependent_steps`:
     - If derived only from the problem statement, assign 'null'.
     - Otherwise, list the minimal, immediately required prior `step_id` values in ascending order, e.g., '[2, 3]'.

3. **Dependency Rules:**
   - Every dependency ID in `direct_dependent_steps` MUST exist in the original set of `step_id` values of the input.
   - Every listed dependency must be a prior step—its `step_id` must be strictly less than the current step (i.e., no self- or future-dependency is allowed).

4. **Structured Output and Consistency:**
   - The number of steps in the structured output MUST match exactly the number of steps in the original input.
   - The output must strictly follow the schema `DependencyGraph`.

**Error Handling**

1. If the input is malformed (e.g., required fields missing, `step_id` missing, or `step_id` values not strictly ascending):

2. Return only a JSON object with an `error` field and a concise message. For example:

   ```
   {"error": "Input data malformed:
   missing step_id in step 3."}
   ```

**Output Format**

Output the structured response with all steps mirroring the original order and count, each annotated with `direct_dependent_steps`. If there is an error, output only the error object as described.

**Demonstration Example**

{Human-written demonstration example}

---

## I Additional Results for Evaluation

The results for final-answer accuracy (PASS@1) and empirical mathematical reasoning ability ($\widehat{\mathcal{R}}$) on three benchmarks across five LLMs are reported in Table 3. The averaged graph-level statistics

Table 3: Final-answer accuracy (PASS@1) and empirical mathematical reasoning ability ($\widehat{\mathcal{R}}$) across three math benchmarks. Parentheses show the gap ($\Delta := \text{PASS@1} - \widehat{\mathcal{R}}$).

| Model | AIME 2025 | | BRUMO 2025 | | HMMT 2025 | |
|---|---|---|---|---|---|---|
| | PASS@1 | $\widehat{\mathcal{R}}$ ($\Delta \downarrow$) | PASS@1 | $\widehat{\mathcal{R}}$ ($\Delta \downarrow$) | PASS@1 | $\widehat{\mathcal{R}}$ ($\Delta \downarrow$) |
| Gemini-2.5-F | 52.4 | 17.0 (35.4↓) | 63.4 | 20.7 (42.7↓) | 38.5 | 5.7 (32.8↓) |
| Gemini-2.5-F-L | 37.4 | 15.9 (21.5↓) | 43.2 | 17.8 (25.4↓) | 28.8 | 7.5 (21.3↓) |
| GPT-4.1 | 26.5 | 16.8 (9.7↓) | 33.3 | 22.8 (10.5↓) | 11.8 | 6.5 (5.3↓) |
| GPT-4.1-M | 30.5 | 20.9 (9.6↓) | 34.4 | 24.6 (9.8↓) | 14.3 | 7.4 (6.9↓) |
| Qwen3-30B | 43.1 | 15.8 (27.3↓) | 46.8 | 21.8 (25.0↓) | 27.3 | 5.6 (21.7↓) |
| Std | 10.3 | 2.1 | 12.2 | 2.5 | 11.0 | 0.9 |

Table 4: Averaged graph-level statistics of sampled DAGs across selected LLMs on AIME 2025.

| Model | Class | #nodes | #edges | density | $d_{in}^{max}$ | $d_{out}^{max}$ |
|---|---|---|---|---|---|---|
| Gemini-2.5-F | All | 32.8 | 48.9 | 11.2% | 4.3 | 7.0 |
| | Incorrect | 35.6 | 53.3 | 10.6% | 4.6 | 8.6 |
| | Correct | 30.2 | 45.1 | 11.6% | 4.1 | 5.5 |
| | Perfect | 23.3 | 30.8 | 13.0% | 3.3 | 3.6 |
| Gemini-2.5-F-L | All | 33.0 | 54.0 | 13.4% | 3.6 | 9.7 |
| | Incorrect | 40.5 | 68.6 | 11.9% | 3.9 | 12.8 |
| | Correct | 21.5 | 31.6 | 15.7% | 3.2 | 4.8 |
| | Perfect | 16.1 | 21.4 | 18.4% | 3.0 | 3.2 |
| GPT-4.1 | All | 17.8 | 21.4 | 16.2% | 2.6 | 3.0 |
| | Incorrect | 18.4 | 22.2 | 15.6% | 2.6 | 3.1 |
| | Correct | 15.9 | 19.3 | 17.5% | 2.7 | 2.9 |
| | Perfect | 14.1 | 16.8 | 19.0% | 2.5 | 2.4 |
| GPT-4.1-M | All | 22.8 | 31.3 | 14.2% | 3.2 | 4.0 |
| | Incorrect | 25.0 | 34.7 | 13.3% | 3.3 | 4.3 |
| | Correct | 17.9 | 23.6 | 16.6% | 3.0 | 3.4 |
| | Perfect | 16.5 | 21.5 | 17.4% | 3.0 | 3.1 |
| Qwen3-30B | All | 21.4 | 31.1 | 16.0% | 3.3 | 4.9 |
| | Incorrect | 23.4 | 34.3 | 14.9% | 3.4 | 5.5 |
| | Correct | 18.9 | 27.2 | 17.2% | 3.1 | 4.1 |
| | Perfect | 14.7 | 19.6 | 20.2% | 2.9 | 3.0 |

for BRUMO 2025 and HMMT 2025 are reported in Tables 5 and 6, respectively. The overall trend is similar to the analysis in Section 5. Additionally, we can obtain the following findings:

- **The change in graph-level statistics is monotonic in $\Delta$.** The variations in the #nodes, #edges, density, and maximum out-degree from **Correct** to **Perfect** increase monotonically with the gap $\Delta$ between PASS@1 and $\widehat{\mathcal{R}}$. When $\Delta$ is small, the statistics of these two classes are nearly identical.

- **Graph-level statistics remain similar across the four classes when raw accuracy is low.** In particular, when PASS@1 is low, their variations across classes are minimal.

## J    DISCUSSION ON FUTURE IMPROVEMENTS

Our proposed metrics can provide a principled foundation for future work aimed at improving both search-based inference and LLM's post-training.

Table 5: Averaged graph-level statistics of sampled DAGs across selected LLMs on BRUMO 2025.

| Model | Class | #nodes | #edges | density | $d_{in}^{max}$ | $d_{out}^{max}$ |
|---|---|---|---|---|---|---|
| Gemini-2.5-F | All | 26.6 | 39.8 | 13.0% | 4.1 | 5.3 |
| | Incorrect | 29.2 | 47.4 | 13.0% | 4.8 | 6.2 |
| | Correct | 25.1 | 35.9 | 13.1% | 3.8 | 4.8 |
| | Perfect | 20.5 | 26.1 | 14.1% | 3.0 | 3.2 |
| Gemini-2.5-F-L | All | 27.9 | 47.7 | 15.5% | 3.8 | 8.3 |
| | Incorrect | 33.2 | 61.5 | 14.5% | 4.0 | 11.1 |
| | Correct | 21.2 | 30.4 | 16.7% | 3.4 | 4.8 |
| | Perfect | 14.1 | 17.4 | 20.1% | 2.6 | 2.9 |
| GPT-4.1 | All | 14.9 | 18.1 | 19.4% | 2.4 | 2.9 |
| | Incorrect | 15.9 | 19.6 | 18.4% | 2.5 | 3.0 |
| | Correct | 13.0 | 15.1 | 21.4% | 2.3 | 2.5 |
| | Perfect | 12.0 | 13.8 | 23.0% | 2.3 | 2.3 |
| GPT-4.1-M | All | 17.1 | 24.1 | 18.7% | 3.0 | 3.7 |
| | Incorrect | 18.6 | 27.2 | 17.7% | 3.1 | 4.1 |
| | Correct | 14.3 | 18.3 | 20.6& | 2.7 | 3.1 |
| | Perfect | 13.1 | 16.6 | 22.1% | 2.7 | 2.9 |
| Qwen3-30B | All | 18.4 | 27.7 | 19.8% | 3.4 | 4.7 |
| | Incorrect | 21.7 | 34.4 | 17.8% | 3.7 | 5.7 |
| | Correct | 14.7 | 20.1 | 22.1% | 3.1 | 3.6 |
| | Perfect | 12.1 | 15.4 | 25.3% | 2.8 | 3.0 |

Table 6: Averaged graph-level statistics of sampled DAGs across selected LLMs on HMMT 2025.

| Model | Class | #nodes | #edges | density | $d_{in}^{max}$ | $d_{out}^{max}$ |
|---|---|---|---|---|---|---|
| Gemini-2.5-F | All | 36.8 | 60.4 | 10.9% | 4.7 | 7.9 |
| | Incorrect | 36.6 | 61.4 | 11.4% | 4.7 | 8.4 |
| | Correct | 37.4 | 59.8 | 10.1% | 4.9 | 7.1 |
| | Perfect | 30.1 | 48.4 | 12.0% | 5.9 | 6.1 |
| Gemini-2.5-F-L | All | 35.8 | 63.0 | 14.0% | 4.1 | 11.2 |
| | Incorrect | 40.4 | 73.3 | 13.4% | 4.1 | 13.1 |
| | Correct | 26.5 | 42.7 | 15.2% | 4.0 | 7.1 |
| | Perfect | 16.8 | 25.9 | 20.6% | 3.9 | 4.2 |
| GPT-4.1 | All | 17.0 | 20.7 | 17.8% | 2.5 | 3.2 |
| | Incorrect | 16.9 | 20.8 | 17.8% | 2.5 | 3.1 |
| | Correct | 17.5 | 20.6 | 17.4% | 2.6 | 3.5 |
| | Perfect | 14.7 | 17.3 | 19.8% | 2.5 | 2.7 |
| GPT-4.1-M | All | 21.1 | 30.2 | 16.0% | 3.0 | 4.2 |
| | Incorrect | 21.1 | 30.2 | 15.9% | 3.0 | 4.1 |
| | Correct | 21.1 | 30.5 | 16.6% | 3.0 | 4.4 |
| | Perfect | 17.2 | 24.3 | 19.4% | 2.9 | 3.6 |
| Qwen3-30B | All | 22.8 | 34.6 | 16.1% | 3.6 | 5.6 |
| | Incorrect | 22.8 | 34.5 | 16.4% | 3.6 | 5.7 |
| | Correct | 22.8 | 35.0 | 15.2% | 3.7 | 5.5 |
| | Perfect | 18.7 | 28.5 | 18.8% | 4.6 | 5.0 |

1. **Guiding Search Policies:** Our framework suggests that logical closeness can serve as a reward to guide graph-based search algorithms like Monte Carlo Tree Search (Pan et al., 2025) or Tree-of-Thoughts (Yao et al., 2023). Instead of relying solely on random branching or value function, search policies could **prioritize** reasoning paths that maintain high

logical coherence. For instance, by favoring branches that close open dependencies or by pruning highly speculative trajectories early. This offers a promising avenue to make search more efficient and goal-directed, steering it toward not just correct but also well-structured solutions.

2. **Informing Reward Design for RL:** The AUC curves inspire a curriculum learning strategy for training reasoning models via RL. One could progressively increase the required level of logical closeness during training, starting from a regime that rewards basic correctness and gradually guiding the model toward the "sweet spot" of perfect reasoning identified by our framework.

## K    EXTENDED EXPERIMENTAL STUDIES

### K.1    STATISTICAL TESTS FOR PASS@1 AND PRR UNDER LARGE SAMPLE

In this section, we aim to establish a statistical test for the difference between PASS@1 and PRR. We extend the sample size from 32 to 128 for `Gemini-2.5-Flash`, `Gemini-2.5-Flash-Lite`, `GPT-4.1`, and `GPT-4.1-mini` on AIME 2025. For mean gap $\Delta$:=PASS@1-PRR, we compute the 95% confidence interval for a binomial proportion using the Wilson score method for each model.

The results are reported in Table 7. All confidence intervals for $\Delta$ exclude zero with tight bounds, confirming with high confidence that the observed gaps between PASS@1 and PRR are statistically significant and substantial in magnitude. Additionally, we note that the statistics from this extended sample align closely with our original results in Table 3.

Table 7: Results of PASS@1, PRR, $\Delta$, and associated confidence interval as CI($\Delta$) across four selected models on AIME 2025.

| **Model** | PASS@1 | PRR | $\Delta$ | CI($\Delta$) |
|---|---|---|---|---|
| Gemini-2.5-F | 53.3% | 17.1% | 36.3% | [34.7%,37.8%] |
| Gemini-2.5-F-L | 38.5% | 15.5% | 22.9% | [21.6%,24.3%] |
| GPT-4.1 | 25.1% | 16.1% | 9.0% | [8.2%,10.0%] |
| GPT-4.1-M | 30.3% | 20.4% | 9.9% | [9.0%,10.9%] |

### K.2    ABLATION STUDIES ON THE SENSITIVITY OF PROMPTS

In this section, we aim to test the sensitivity of few-shot prompt in Appendix F via perturbing the prompts in the following two ways:

- Convert nested bullet points into a structured table format, see details in Appendix G.1.
- Rephrase every sentence with synonyms and alternative grammatical structures, see details in Appendix G.2.

We select `GPT-4.1` as test model and AIME 2025 as test dataset. To rigorously assess prompt sensitivity, we employ the TOST (Two One-Sided Tests) procedure for statistical equivalence testing on five graph statistics introduced in Section 4.2. We consider two-sample mean difference per-problem then set the significance level to $0.05$ and equivalence margin to $0.10$. The testing results are reported in Table 8 for re-format and Table 9 for rephrase. Fig. 8 reports AUC scores on DAG-MATH formatted CoT with original and two perturbed prompts, see Table 10 for more details. We have the following observations:

- **Structural Consistency:** From Table 8 and Table 9, our equivalence testing demonstrates that the DAG structures generated under different prompt variations are statistically equivalent across all five graph-level statistics.
- **Robust Evaluation:** In Fig. 8 and Table 10, we can observe that the variation in PRR/AUC is minimal, suggesting our metrics capture inherent model reasoning capabilities rather than artifacts of specific prompting strategies.

Table 8: TOST equivalence testing between original and re-format prompts across 30 problems (per-problem paired mean aggregation). CI denotes 90% confidence interval. $p_{\text{left}}$ and $p_{\text{right}}$ are the two one-sided p-values one for lower and upper bound test.

| Stat. | Mean Diff. | CI | $p_{\text{left}}$ | $p_{\text{right}}$ | Equivalent |
|-------|-----------|-----|-------|-------|-----------|
| #nodes | 0.56 | [0.18, 0.95] | $2.59 \times 10^{-11}$ | $7.76 \times 10^{-6}$ | YES |
| #edges | 0.44 | [-0.11, 0.99] | $5.01 \times 10^{-9}$ | $7.49 \times 10^{-6}$ | YES |
| density | -0.23% | [-0.37%, -0.09%] | $7.99 \times 10^{-8}$ | $2.22 \times 10^{-13}$ | YES |
| $d_{\text{in}}^{\text{max}}$ | 0.03 | [-0.04, 0.10] | $5.64 \times 10^{-8}$ | $2.81 \times 10^{-6}$ | YES |
| $d_{\text{out}}^{\text{max}}$ | 0.01 | [-0.09, 0.12] | $1.16 \times 10^{-5}$ | $4.18 \times 10^{-5}$ | YES |

Table 9: TOST equivalence testing between original and re-phrase prompts across 30 problems (per-problem paired mean aggregation). CI denotes 90% confidence interval. $p_{\text{left}}$ and $p_{\text{right}}$ are the two one-sided p-values one for lower and upper bound test.

| Stat. | Mean Diff. | CI | $p_{\text{left}}$ | $p_{\text{right}}$ | Equivalent |
|-------|-----------|-----|-------|-------|-----------|
| #nodes | 0.89 | [0.52, 1.26] | $3.83 \times 10^{-13}$ | $2.82 \times 10^{-4}$ | YES |
| #edges | 0.82 | [0.30, 1.35] | $9.80 \times 10^{-11}$ | $1.29 \times 10^{-4}$ | YES |
| density | -0.3% | [-0.49%, -0.17%] | $5.30 \times 10^{-6}$ | $2.34 \times 10^{-13}$ | YES |
| $d_{\text{in}}^{\text{max}}$ | 0.03 | [-0.04, 0.10] | $2.58 \times 10^{-8}$ | $1.89 \times 10^{-6}$ | YES |
| $d_{\text{out}}^{\text{max}}$ | 0.007 | [-0.10, 0.11] | $1.40 \times 10^{-5}$ | $2.67 \times 10^{-5}$ | YES |

### K.3 ABLATION STUDIES ON FORMATTING CONSTRAINT AND CIRCULARITY

In this section, we conduct ablation studies in two aspects:

- analyze the impact of formatting constraint comparing to natural CoT;

- examine the consistency of DAG constructions by different model families for evaluation.

We design two prompting methods to enable DAG parsing by external thinking/non-thinking model, which makes our PPR/AUC metric apply to any step-by-step CoT without formatting. We select `GPT-4.1` as test model and AIME 2025 as test dataset.

**Natural CoT Generation:** We compare DAG-MATH formatted CoT with natural CoT to check whether the formatting constraint degrades the reasoning ability in natural form or not. We design a minimal zero-shot prompt for natural CoT which only requires the model to generate step-by-step solution and write the final answer in boxed format for convenience of evaluation. Then, we independently draw 32 natural CoT samples for each problem.

**DAG Parsing:** Since the prompting evaluation in Section 5 requires self-parsing for the dependency structure, we aim to show there are no circular dependency by adding dependencies for both formatted and natural CoT via external LLMs. Besides, the corresponding parsing method can annotate DAG for natural CoT, which generalizes the utility of our PPR/AUC metric. We evaluate three modes:

- parsing DAG by `Gemini-2.5-Flash-Thinking`,

- parsing DAG by `GPT-o4-mini`,

- self-parsing.

For thinking LLMs, since the construction of dependencies can be performed within thinking process, we design a zero-shot prompting method to provide instructions for model to annotate DAGs, see Appendix H.1 for details. Since `GPT-4.1` is a non-thinking LLM (instant answer), we add a `reason` field to analyze dependencies before generating `direct_dependent_steps` field, in order to enable self-parsing for natural CoT. Additionally, we add one human-written demonstration example for better instruction following. See Appendix H.2 for more details.

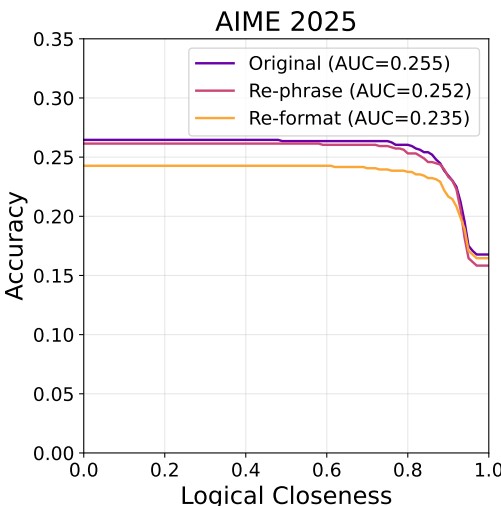

Figure 8: The AUC curves of averaged accuracy under the logical closeness rate for DAG-MATH formatted CoT under original, re-format, and re-phrase prompts.

Table 10: The results of PASS@1, PRR, and AUC values for DAG-MATH formatted CoT under original, re-format, and re-phrase prompts.

|  | Original | Re-format | Re-frame |
|---|---|---|---|
| PASS@1 | 26.5% | 24.3% | 26.1% |
| PRR | 16.8% | 16.5% | 15.8% |
| AUC | 25.5% | 23.5% | 25.2% |

**Results** Fig. 9 reports AUC scores on natural CoT and DAG-MATH across different parsing models (see more details in Table 11). We perform the same TOST as Appendix K.2 with margin 0.25 and all statistics are tested as equivalent between self-parsing and external parsing. We have the following findings.

- **Utility on Natural CoT & Format Dependency:** Our PRR/AUC metrics remain highly informative even when applied to free-form natural CoT. From Fig. 9, the AUC metric clearly shows that the natural CoT can achieve more than 70% logic-closed nodes, which is consistent across different LLMs. The AUC values for natural CoT are comparable to those for DAG-MATH, demonstrating that the metrics capture fundamental reasoning patterns regardless of output format. In Table 11, the consistently higher PRR for DAG-MATH indicates that the structured format enhances logical coherence by encouraging more *dependency-aware reasoning*. This confirms that the DAG-MATH format amplifies rather than creates the signal of logical coherence.

- **No Circular Dependency:** The consistency of AUC scores across different parsing methods-ranging from 23.8% to 24.7% for natural CoT and 24.9% to 25.5% for DAG-MATH-directly addresses circularity concerns. This consistency is further enhanced by our TOST which all achieve statistical equivalence. This shows that logical closeness is an objective property that can be reliably measured by independent models, validating the robustness of our evaluation framework. We randomly draw 10 samples which are closed by self-parsing but unclosed by others then evaluate the dependency structure manually by human. Surprisingly, we find that self-parsing works better than external parsing, which matches the fact that LLM generates CoT following the internal dependency studied in Ye et al. (2025). There might be an extra understanding stage needed when parsing with external models.

- **Format Constraint Impact:** The DAG-MATH format does not degrade natural reasoning flexibility, as evidenced by better PASS@1 scores (26.5% for DAG-MATH vs. 25.7% for natural CoT). The significantly lower average node count and standard deviation in DAG-MATH indi-

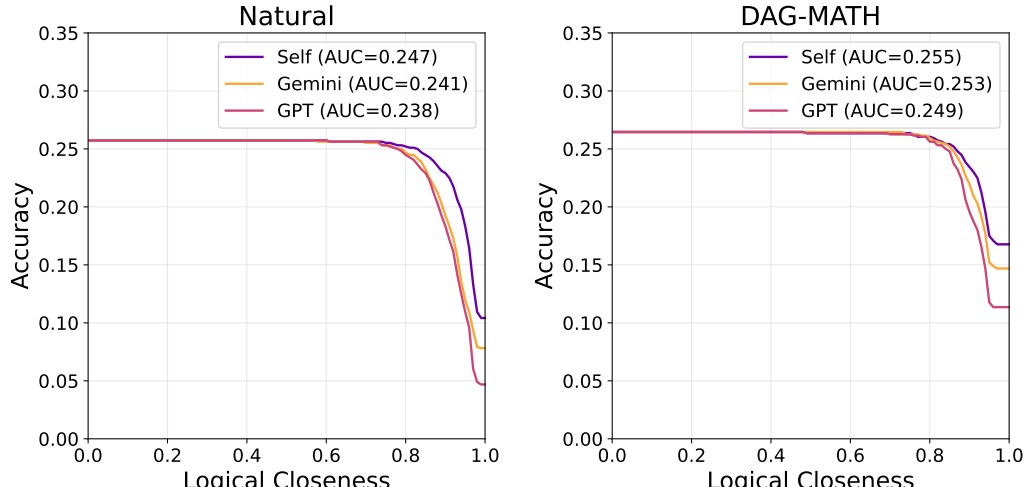

Figure 9: The AUC curves of averaged accuracy under the logical closeness rate for natural CoT and DAG-MATH across self-parsing, parsing by `Gemini-2.5-Flash-Thinking` (Gemini for short), and parsing by `GPT-o4-mini` (GPT for short).

Table 11: The results of average number of nodes (#Nodes), PASS@1, PRR, AUC scores for natural CoT and DAG-MATH across self-parsing, parsing by `Gemini-2.5-Flash-Thinking` (Gemini for short), and parsing by `GPT-o4-mini` (GPT for short).

| Metric | Natural | | | DAG-MATH | | |
|---|---|---|---|---|---|---|
| #Nodes | $40.3_{\pm 26.6}$ | | | $17.8_{\pm 7.0}$ | | |
| PASS@1 | 25.7% | | | 26.5% | | |
| | Self | Gemini | GPT | Self | Gemini | GPT |
| PRR | 10.4% | 7.8% | 4.7% | 16.8% | 14.7% | 11.4% |
| AUC | 24.7% | 24.1% | 23.8% | 25.5% | 25.3% | 24.9% |

cates the format encourages more focused reasoning without sacrificing accuracy. The higher PRR across all parsing methods for DAG-MATH suggests the format actually enhances reasoning quality by promoting logically closed derivations, while natural CoT causes extra verification burden due to a large amount of irrelevant information.

### K.4 ABLATION STUDIES ON CROSS-FAMILY FEW-SHOT EXAMPLES

In this section, we conduct ablation studies regarding model bias in few-shot examples. We fix the same few-shot example problems and repeat the gold-standard DAG construction procedure introduced in Appendix D by `Gemini-2.5-Pro`, different to `GPT-o4-mini` used in benchmark construction. Next, we select `GPT-4.1` and `Gemini-2.5-Flash` to generate DAG-MATH samples under prompts with few-shot examples from two different model families, which forms a cross-family comparison. Fig. 10 shows the AUC curves with more details in Table 12. The results show that while there is a slight performance drop when using examples from a different family, the effect is modest and does not fundamentally alter the relative rankings or conclusions drawn from our metrics. Furthermore, we perform same TOST as Appendix K.2 to test the equivalence of DAG structures under cross-family examples. `GPT-4.1` achieves statistical equivalence at a margin of 0.1 and `Gemini-2.5-Pro` at 0.25, which is a strong evidence.

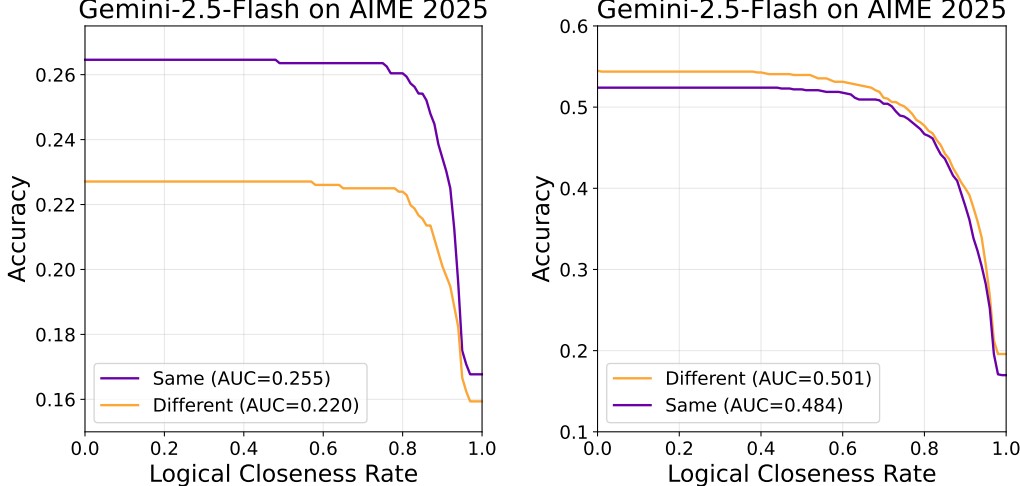

Figure 10: The AUC curves of averaged accuracy under the logical closeness rate for DAG-MATH formatted CoT under prompts with few-shot examples from same vs different model family.

Table 12: Results of PASS@1 and PRR for two test models with few-shot examples from same/different model family on AIME 2025.

|  | PASS@1 | | PRR | |
|---|---|---|---|---|
| **Family** | same | diff. | same | diff. |
| GPT-4.1 | 26.5% | 22.7% | 16.8% | 15.9% |
| Gemini-2.5-Flash | 54.4% | 52.4% | 19.6% | 17.0% |

## L    THE USE OF LARGE LANGUAGE MODELS (LLMS) STATEMENT

We claim that LLMs were used minimally for language polishing. For example, we use ChatGPT to make the abstract more precise and concise, which has no effect on the technical contribution of this paper.

