# OpenReview forum: "DAG-Math: Graph-of-Thought Guided Mathematical Reasoning in LLMs"
_ICLR.cc/2026/Conference — ICLR 2026 Poster_

### Official Review · Reviewer_GdSj · 2025-10-27

**Soundness:** 2
**Presentation:** 3
**Contribution:** 2
**Rating:** 6
**Confidence:** 2

**Summary:**

This paper proposes a DAG-based framework for representing CoT reasoning and introduces the concept of logical closeness, enabling fine-grained evaluation of LLM mathematical reasoning beyond final-answer accuracy. Its approach assesses the coherence and consistency of logical dependencies along a CoT trajectory, rather than focusing solely on whether the solution is correct. Furthermore, the authors construct a benchmark of DAG-formatted mathematical problems derived from existing datasets and provide empirical analyses linking graph-level characteristics—such as size, density, and branching complexity—to problem difficulty.

**Strengths:**

- This work formalizes the notion of logical closeness and proposes a metric, the perfect reasoning rate, based on this notion to measure LLMs' logical consistency beyond the final output. This indeed addresses an important question of whether an LLM arrives at a correct answer through genuine logical reasoning or mere pattern matching.
- The formalization in Sections 2 and 3 is clearly presented and easy to follow.
- Section 5 and Appendix B offer several interesting insights, such as the correlation between graph structure and problem difficulty, and how a correct final answer may still arise from unclosed or flawed reasoning.

**Weaknesses:**

- The DAG-MATH benchmark presented in the paper is validated using symbolic correctness and an LLM-as-Judge approach. I assume that SymPy is employed to verify mathematical equivalence, while the logical dependencies between nodes (i.e., whether an edge should exist) are assessed by the LLM-as-Judge. However, as the paper itself demonstrates, LLMs can often produce superficially consistent but logically inconsistent reasoning. While using LLMs for judgment is a practical solution, it would be helpful for the authors to further justify the reliability of this dataset construction and evaluation methodology.
- Building on this point, reliable automation of the logical closeness check appears challenging, if not infeasible, since formalizing a DAG from natural-language CoT inherently involves subjective interpretation, particularly for more complex problems. For instance, reasonable disagreement could arise over whether a given edge should connect two specific nodes.

**Questions:**

- How can we trust an LLM-as-Judge to reliably evaluate the logical coherence of DAG constructions, particularly when edges are intended to represent valid inference paths? Have the authors conducted any analyses or validation studies to assess the consistency or accuracy of these judgments?
- At first glance, Figure 4 being a line plot was somewhat confusing. Do the authors plot accuracy against varying levels of logical correctness rates and then smooth the resulting curve? A brief clarification in the caption or text might help.
- I also wonder whether different node segmentation choices could exist for the same reasoning trajectory. If so, how sensitive are the proposed approach and the PRR metric to such segmentation differences?

---

> ### Author Response · Authors · 2025-11-21
> **Response to Reviewer GdSj**
>
> We deeply appraciate reveiwer's efforts and insightful comments.
>
> First, we invite reviewer to Common Reply 1 for a series of robustness experiments with analysis on:
> - *reliability of evaluation methodology*
>
> Common Reply 2 for details of:
> - *reliability of this dataset construction and human evaluation of benchmark*
>
> and Common Reply 5 for a detailed discussion on:
> - *reliable automation of logical closeness check*
>
> We revise our Symbolic & LLM-as-judge validation procedure of benchmark construction in **Appendix D** of the updated version to provide a better declaration.
>
> Next, we address reviewer's specific concerns point by point below:
>
> **Q1** *clarification on Figure 4*
>
> **A1** Figure 4 plots the model's **final-answer accuracy** against a varying **minimum threshold** for the Logical Closeness Rate.
>
> To plot each curve, we first calculate the Logical Closeness Rate for every generated DAG (defined as `#Closed Nodes / #Total Nodes`). We then plot the accuracy of the model's final answer on the subset of problems where the DAG's Logical Closeness Rate meets or exceeds the given threshold. This creates a curve showing how accuracy degrades as we demand more logically coherent reasoning traces. The resulting curve is **not smoothed**: it is a direct plot of the computed accuracies at a fine-grained grids of threshold (0.01 increment from 0 to 1). We updated the caption of **Figure 4** in the revised manuscript.
>
> **Q2** *different node segmentation choices, robustness of PRR*
>
> **A2** We acknowledge that the same step has different segmentation choices such as varying granularity. However, our framework is designed to be robust to such variations for two key reasons:
> 1. By prompting models to generate high-granularity steps to standardize and, crucially, by having them **explicitly cite parent steps via their step IDs**, we ensure that the core dependency topology is preserved. Whether a derivation is presented as one coarse step or two finer steps, the cited ancestor relationships will consistently reflect the same underlying logical flow.
> 2. The PRR metric is an expected value averaged over a pool of DAG samples, which naturally smooths out variations in node segmentation. This makes PRR a robust **population-level** measure of reasoning fidelity—analogous to how a **graphon** captures essential graph structure—reflecting a model's overall tendency toward perfect reasoning.

---

> ### Author Response · Authors · 2025-11-26
>
> Dear Reviewer GdSj,
>
> Thank you again for your constructive reviews. As the discussion window will end in one week, we would like to gently invite you to revise our response and let us know if it satisfactorily addresses your comments. Please feel free to request any further clarification.
>
> Kind Regards
>
> Authors

---

### Official Review · Reviewer_61B1 · 2025-11-01

**Soundness:** 4
**Presentation:** 3
**Contribution:** 3
**Rating:** 8
**Confidence:** 3

**Summary:**

The paper proposes to model mathematical reasoning’s CoT traces as a rule-based stochastic process over task-specific DAGs, where nodes represent reasoning states and edges encode inference rules or justifications. Building on this formulation, the paper introduces logical closeness as a new metric to evaluate the model's reasoning trajectory. It also presents a benchmark of gold-standard DAG-MATH graphs with verified logical structures and statistical analyses relating graph properties to problem difficulty. Finally, the paper evaluates several large language models by prompting them to produce formatted CoT reasoning on mathematical benchmarks and examines how their reasoning abilities correlate with the proposed DAG-MATH framework.

**Strengths:**

- The paper is clearly written and well-organized.
- The idea of representing CoT reasoning with DAG-MATH is interesting and novel. The proposed metrics are also new and conceptually sound.
- The empirical results are informative, showing how graph structures reflect problem difficulty and reasoning quality.

**Weaknesses:**

- Enforcing the DAG-MATH format may degrade the natural reasoning flexibility of LLMs. It would help to include an analysis or ablation comparing performance with and without this formatting constraint. Furthermore, if the few-shot examples are drawn from a specific model family, models of the same family might have an advantage because their reasoning patterns are similar.
- The analysis is primarily quantitative. Some qualitative examples or case studies of the generated DAG-MATH graphs, especially highlighting common reasoning errors or structural failures, would strengthen the insights.
- In Section 2.2, the paper mentions that thinking LLMs can be viewed as “an exploration of the task-specific DAG with self-correction or backtracking, but its final output … is still consistent with our transition rule.”, but empirical results did not include reasoning models, which could strengthen the value of the paper.
- DAG-MATH has limited scalability. Complex problems with very long and multiple reasoning traces are computationally expensive to construct. Those with cyclic reasoning or backtracking are hard to capture in a strictly acyclic form.

**Questions:**

- How exactly is the branching of reasoning paths determined?
- Since the canonicalization turns reasoning steps into SNF, does it limit the type of mathematical questions that DAG-MATH can apply to?
- In lines 63-64, what does it mean that the other works “fail to capture long-range and cross-branch dependencies, as well as the goal-directed, absorbing-state nature of CoT”?

---

> ### Author Response · Authors · 2025-11-21
> **Response to Reviewer 61B1 [1/2]**
>
> We greatly thank the reviewer's efforts and constructive comments with positive support.
>
> First, we invite reveiwer to Common Reply 1 for detailed discussions on:
> - *with/without formatting constraint*
> - *potential advantage of few-shot examples from same family*
>
> and Common Reply 3 for a detailed discussion on:
> - *scalability of DAG-MATH, rationale of acyclicity assumption*
>
> Next, we address reviewer's specific concerns point by point below:
>
> **Q1** *qualitative examples/case studies*
>
> **A1** One case study was given in **Appendix B** in our original submission. We discuss it here for better readability.
>
> In Appendix B, we provide a complete analysis of a complex geometry problem from AIME 2025. We analyze four correct-but-unclosed CoT trajectories from ```Gemini-2.5-Flash``` by human. The case study qualitatively reveals:
>
> - **Strategy-Switching Errors:** We show how the model begins a "reflection-swap" strategy but then abandons it after mid-derivation to use a "shoelace formula" strategy, leaving crucial nodes from the first strategy unclosed. This pattern occurs across three trajectories, with one particularly long trajectory exhibiting even severer fragmentation—switching between four distinct strategies.
>
> - **Specific Unclosed Nodes:** There are two common unclosed nodes, which are property-declaritive (collinearity), are not cited. We identify that some subsequent steps implicitly use to add segment lengths but the LLM does not recognize.
>
> **Q2** *empirical results for thinking LLMs*
>
> **A2** We add experiments on thinking LLMs in **Section 5.2**. We find that thinking mode significantly boosts both PASS@1 and PRR, the substantial performance gap between two metrics persists (35.4% w vs. 28.0% w/o for Flash, 21.5% w vs. 24.7% w/o for Flash-Lite). The continued gap indicates that thinking enhances the model's exploration of the task-specific DAG, yielding more correct and logically closed trajectories, but does not eliminate the tendency on search over logical coherence.
>
> **Q3** *branching of reasoning paths*
>
> **A3** In our framework, branching can be determined through both qualitative patterns and quantitative graph metrics:
>
> 1.  **Common Branching Patterns:** We empirically identify that branching primarily occurs in two common scenarios:
>     *   **Task Decomposition:** A complex node (e.g., "calculate the total area") is decomposed into several semi-independent sub-tasks or lemmas (e.g., "calculate area of part A," "calculate area of part B"). These sub-derivations form distinct branches that are later aggregated.
>     *   **Pivotal Inference:** A single key node serves as a premise for multiple, parallel derivations. For instance, establishing a core monthly-spent equation may then be used to derive the individual expenses for different people in parallel, creating several reasoning branches from a single parent node.
>
> 2.  **Empirical Quantification via Graph Statistics:** We formally identify and measure the intensity of branching using graph-level statistics. Specifically, the **maximum out-degree** in the DAG serves as our primary metric for branching complexity. As detailed in Section 4.2 and Figure 3, we find that harder problems systematically induce DAGs with higher maximum out-degree, indicating that complexity scales more through branching into multiple inferences from a single step than through deep, linear chains.
>
> **Q4** *canonicalization limits problem type*
>
> **A4** The mapping from natural-language steps to SNF is fundamentally analogous to how mathematical statements are formalized in systems like Lean. Therefore, by type theory, rather than limiting the type of questions, canonicalization generalizes the problem scope.
>
> This process aims to abstract away superficial linguistic variations (synonyms, phrasing, variable names) to expose the core mathematical semantics, just as Lean's kernel operates on normalized code regardless of their human-readable presentation. Consider the concept of "defining a variable." An LLM might generate the following different sentences, all of which are logically equivalent:
> *   "Let $x$ represent the number of apples Alice has."
> *   "Denote the quantity of Alice's apples by $x$."
> *   "We define $x$ to be Alice's apple count."
> *   "Set $x$ equal to the number of apples belonging to Alice."
>
> Despite their different phrasing, we can use Lean syntax `def x : Nat := Alice.apples` to represent them all.

---

> ### Author Response · Authors · 2025-11-21
> **Response to Reviewer 61B1 [2/2]**
>
> **Q5** *lines 63-64: meaning of other works on "“fail to capture long-range ..."*
>
> **A5** We give a detailed explanation here to avoid confusion:
> 1.  **Failure to capture long-range and cross-branch dependencies:** The framework in [1] assume a **Markovian** structure where each reasoning step depends only on the previous one, along with a **recurrent** logical ordering that is invalid in the context of mathematical reasoning. This ignores that a step may depend on a conclusion from much earlier in the chain, or may need to combine results from two separate, parallel branches of reasoning (cross-branch dependencies). The framework in [2] that restrict paths to linear chains is similarly limited.
>
> 2.  **Failure to capture the goal-directed, absorbing-state nature:** The "absorbing-state" refers to the final answer, which terminates the reasoning process. In contrast, model based on recurrent and reversible Markov chains ([1]) lack a dedicated termination, making the process more akin to an open-ended exploration than a **goal-directed** generation for a solution. Furthermore, modeling CoT as a fixed, deterministic DAG of only the correct path ([3]) fails to capture the stochastic sampling that LLMs perform, which can include branching into incorrect paths before arriving at the answer.
>
> Our framework integrates an task-specific DAG (as a search space containing multiple paths to correct/incorrect answer) with a rule-based stochastic process to directly model these crucial aspects: it natively supports **long-range and cross-branch dependencies** via the DAG structure, and explicitly incorporates **goal-directedness** through its absorbing sink nodes and transition rules that govern the sampling of solution trajectories. We will revise this part in the updated version to explain these distinctions more carefully.
>
> **Reference:**
>
> [1] Kim, Juno, et al. "Metastable Dynamics of Chain-of-Thought Reasoning: Provable Benefits of Search, RL and Distillation." ICML 2025.
>
> [2] Shalev-Shwartz, Shai, and Amnon Shashua. "From Reasoning to Super-Intelligence: A Search-Theoretic Perspective." arXiv (2025).
>
> [3] Dziri, Nouha, et al. "Faith and fate: Limits of transformers on compositionality." NeurIPS 2023.

---

> ### Author Response · Authors · 2025-11-26
>
> Dear Reviewer 61B1,
>
> We appreciate your thoughtful comments and positive support. With the discussion phase concluding in one week, we would be grateful if you could review our response and advise whether it resolves the issues you raised. We are happy to elaborate further if needed.
>
> Kind Regards
>
> Authors

---

### Official Review · Reviewer_utCQ · 2025-11-05

**Soundness:** 2
**Presentation:** 3
**Contribution:** 3
**Rating:** 6
**Confidence:** 3

**Summary:**

This paper introduces DAG-MATH, which is a framework to evaluate mathematical reasoning of LLMs through modeling Chain-of-Thought as stochastic process over directed acyclic graphs. The authors propose metric called "logical closeness" for distinguishing between when model solves problem by search versus real logical inference. They make benchmark with 2,894 gold-standard DAGs and test five LLMs, discovering that although PASS@1 scores vary much between models, their perfect reasoning rates stay quite similar, which suggests search is inflating the accuracy metrics.

**Strengths:**

The paper tackles really important problem about understanding whether LLMs achieve correct answers through systematic search or through genuine logical reasoning, which is fundamental question for the field. The DAG-based formalization is quite novel approach that sits nicely between completely free-form CoT and very formal systems like LEAN verification, making it more practical to use. The logical closeness metric gives us insights that go beyond simple PASS@k metrics that everyone uses.

The empirical analysis is quite comprehensive, showing interesting patterns about how DAG statistics like number of nodes, edges, density and branching correlate with problem difficulty. It reveals that harder problems create larger and more sparse graphs with higher branching, which makes sense intuitively. The finding that search and exploration inflate PASS@1 while actual reasoning ability measured by PRR stays comparable across different models is really actionable insight that changes how we should think about evaluating these systems. Also good that authors released the benchmark and code for others to use.

**Weaknesses:**

There is concerning circularity in how the benchmark was constructed - using GPT-4 and Qwen to create the "gold standard" DAGs means the benchmark is essentially created by same type of models that are being evaluated, which introduces obvious biases. The theoretical justification feels not enough developed. Why should we believe this specific DAG formalization captures what "true" reasoning means? The stochastic process described in Equation 1 seems somewhat arbitrary choice without proper justification.

The statistical analysis lacks rigor with only 32 samples per problem and no significance testing provided for the differences claimed between PASS@1 and PRR. There is no analysis about robustness - what happens when same problem can be formulated with different but equivalent DAG structures? The scope is quite limited, restricting to mathematics problems with difficulty below 6, and missing comparisons with other approaches like process reward models that also try to do step-level verification. The three-stage prompting methodology might be imposing particular reasoning patterns that are not universal. Most importantly, there is no human validation beyond what the models themselves produce, which is problematic for claiming these are "gold standard" solutions.

**Questions:**

How did you validate that the DAGs generated by GPT and Qwen actually represent correct reasoning structures? It seems crucial to have human experts verify at least subset of these DAGs to ensure the benchmark quality. Many mathematical problems have multiple valid solution approaches - how does the logical closeness metric handle cases where completely different but valid DAG structures could exist for same problem?

Could you provide proper statistical significance tests for the differences you claim between PASS@1 and PRR? How sensitive are all these results to the specific prompting strategies you used? If you changed the prompts slightly, would the DAG structures and evaluation results change significantly?

What is the practical path from these evaluation insights to actually improving LLM reasoning capabilities? The paper identifies interesting patterns but doesn't suggest how to use this knowledge for making better models. How does this framework compare empirically with recent work on process reward models from OpenAI and others that also try to verify reasoning at step level?

Why did you choose to require exactly one assertion per node - this seems quite restrictive and arbitrary? And why make logical closeness binary measure instead of having gradient that could capture partial correctness better? Finally, have you considered that the models might be following completely different internal reasoning process and the DAG structure is just post-hoc rationalization that we impose on their outputs?

---

> ### Author Response · Authors · 2025-11-21
> **Response to Reviewer utCQ [1/2]**
>
> We deeply appreciate the reviewer’s efforts and the positive support.
>
> We invite reviewer to Common Reply 1 for detailed discussions on:
> - *model bias of evaluation*
> - *statistical significance tests for the differences between PASS@1 and PRR*
> - *sensitivity of prompting strategy*
>
> Common Reply 2 for details of:
> - *human evaluation of our gold-srandard benchmark*
>
> and Common Reply 4 for:
> - *potential improvement from evaluation insights*
>
> Next, we address reveiwer's specific concerns point-by-point below:
>
> **Q1** *theoretical justification, DAG formalization, stochastic process in Eq.(1), one-assertion-per-node design*
>
> **A1** The DAG framework is not to provide a definition of reasoning but to establish a rigorous pratical measurement of reasoning ability through three necessary conditions derived from mathematical practice: 1) identify all required premises, 2) conduct correct logical inference from premises to reach the conclusion, and 3) perform accurate primitive operation (line 39-40). Our DAG framework operationalizes these conditions in a concrete, verifiable way by defining **Nodes** as identified premises/intermediate conclusions and **Edges** as logical dependencies.
>
> Eq. (1) emerges naturally from this perspective as a topology-respecting traversal constraint which requires nodes to be generated only after their logical prerequisites are satisfied, thereby capturing the structural validity of reasoning. Models with transition probabilities violate this rule does not engage in coherent reasoning and merely produces unordered assertions. Based on the DAG in Figure 1, an example trajectory violates Eq. (1) could be $(v_1,v_2,v_3,v_4,v_5,\boxed{v_7,v_6},v_8,v_{10})$. The model derives the solutions to quadratic equation **before** setting up the equation (unknown), which is logically impossible.
>
> The one-assertion-per-node design is a normalization step to ensure consistent granularity for a particular focus on DAG topology, not an arbitrary restriction. This creates a **standard** for comparing logical flow across different models and reasoning styles, rather than being confounded by variations in step bundling.
>
> **Q2** *Robustness-same problem with different but equivalent DAGs, problems with multiple valid solution*
>
> **A2** The stuctural difference of equivalent DAGs mainly comes from varying step granularity. The PRR metric is an expected value of logical closeness over population, which smooths out variations in granularity, analogous to the **graphons** that are close in the cut distance, as they preserve the essential dependency topology.
>
> Furthermore, DAG-MATH ensures robustness by explicitly encoding dependencies via **step-ID** citations, which decouple evaluation from superficial step phrasing. Logical Closeness is then assessed based on whether all nodes contribute to the final answer through these cited dependencies, not on fixed graph. This approach guarantees that both equivalent DAGs and DAGs of alternative solution are recognized as valid.
>
> **Q3** *limited scope: problems difficulty below 6*
>
> **A3** This setting is to ensure the reliability of benchmark. We identified difficulty 6 as a critical transition point: problems within 6 maintain a feasible success rate for generating valid gold-standard DAGs, while no problems above this threshold could be solved because these problems reach IMO-level complexity. Our framework is general and can be extended to harder problems as model capabilities improve.
>
> **Q4** *comparisons with methods on step-level verification, e.g., PRMs*
>
> **A4** The goal of ours differs from PRMs. PRMs excel at verifying the stepwise correctness but **cannot** diagnose the global logical structure, which shown by [1]. Our DAG-MATH framework is not a step-correctness verifier; it is a structural analyzer that assesses whether a sequence of steps forms a coherent, logically closed derivation. This focus on dependency topology and reasoning fidelity is orthogonal to the value of PRMs.
>
> **Q5** *three-stage prompting imposes particular reasoning patterns*
>
> **A5** Our three-stage methodology is carefully designed to separate reasoning generation from structural formalization, ensuring it does not impose specific patterns.
>
> Stage 1 prompts only for a high-granularity (can improve CoT's quality [2,3,4]), step-by-step CoT that does not prescribe a specific logical path. The core reasoning pattern emerges naturally from the LLM here. Stages 2 and 3 are purely descriptive: they merely annotate the dependencies (Parents) and justifications (Edges) for the already-fixed sequence of Nodes. This process extracts the latent logical structure without altering the inherent reasoning flow. Therefore, the resulting DAG is a formalization of the model's own reasoning, not an imposition of an external template.

---

> ### Author Response · Authors · 2025-11-21
> **Response to Reviewer utCQ [2/2]**
>
> **Q6** *logical closeness, binary measure*
>
> **A6** Our binary definition of Logical Closeness is used to rigorously define **Perfect Reasoning**. Besides, we explicitly address partial correctness through the **Logical Closeness Rate** and **AUC curves** (Figure 4). We directly plot the accuracies at a fine-grained grids of threshold (0.01 increment from 0 to 1) to capture how accuracy evolves with reasoning coherence. Thus, our framework inherently incorporates the gradient measure the reviewer suggests.
>
> **Q7** *different internal reasoning process, post-hoc rationalization*
>
> **A7** We appreciate the reviewer's thoughtful concern on the post-hoc rationalization. Recent work [5] shows LLM's outputs are **consistent** with the internal dependencies pre-computed *in weight* before generating a solution, which aligns directly with the core principles of our DAG framework. Our framework makes this latent reasoning process explicit and evaluable, not imposed post-hoc.
>
> Our work builds upon this insight but makes contributions in rigorous mathematical formulation and practical generality. While probing in [5] requires direct access to a model's internal states (open-sourced weights) and the design of specific classifiers to be trained, our DAG-MATH framework operates solely on the model's final output, which makes our method model-agnostic.
>
> **Reference:**
>
> [1] Xu, Yuhui, et al. "Reward Models Identify Consistency, Not Causality." arXiv (2025).
>
> [2] Berchansky, Moshe, et al. "CoTAR: Chain-of-Thought Attribution Reasoning with Multi-level Granularity." EMNLP (Findings). 2024.
>
> [3] Chen, Xinghao, et al. "Unveiling the key factors for distilling chain-of-thought reasoning." ACL (Findings). 2025.
>
> [4] Wang, Ru, et al. "Beyond in-distribution success: Scaling curves of cot granularity for language model generalization." arXiv (2025).
>
> [5] Ye, Tian, et al. "Physics of Language Models: Part 2.1, Grade-School Math and the Hidden Reasoning Process." ICLR 2025.

---

> ### Author Response · Authors · 2025-11-26
>
> Dear Reviewer utCQ,
>
> We are grateful for your insightful reviews. Since the discussion period is scheduled to close in one week, we kindly invite you to verify that our response resolves your concerns. We would be pleased to provide any additional detail you need.
>
> Kind Regards
>
> Authors

---

### Official Review · Reviewer_wXKz · 2025-11-05

**Soundness:** 2
**Presentation:** 3
**Contribution:** 2
**Rating:** 4
**Confidence:** 4

**Summary:**

This paper introduces DAG-MATH, a framework designed to formalize and evaluate the Chain-of-Thought (CoT) trajectories generated by Large Language Models (LLMs) in mathematical reasoning. The CoT is modeled as a rule-based stochastic process over a Directed Acyclic Graph (DAG), where nodes are intermediate states. The core proposal is the Logical Closeness metric, which quantifies the fidelity of an LLM's path against a "gold standard" DAG, yielding the Perfect Reasoning Rate (PRR) and AUC scores. The authors claim this provides a superior diagnostic tool compared to simple final-answer metrics like PASS@k. The resulting DAG-MATH benchmark is built using LLM-generated structured outputs.

**Strengths:**

1. The underlying idea of treating CoT as a DAG traversal is fundamentally sound and offers a pathway for structured reasoning analysis beyond token-level checks. This is the paper's primary and most important strength.
2. The authors have created impressive few-shot prompts to enforce their complex output format, which is a valuable demonstration of structured generation control in LLMs. The visual examples of the DAGs are convincing.
3. The metric correctly isolates failure modes like speculative branching and imperfect reasoning, which are invisible to simple PASS@k.

**Weaknesses:**

1. The PRR/AUC metric confuses adherence to the authors' custom template with true logical reasoning ability. The paper must provide evidence that this metric holds up when applied to non-formatted, naturally generated CoT.
2. A critical omission is the lack of comparison with or contextualization against MCTS or similar graph-based search methods. If the goal is to improve reasoning, how does the DAG-MATH diagnosis inform or relate to these established LLM search strategies?
3. The use of LLMs to generate the ground truth DAGs for their own evaluation introduces a circular dependency. This casts significant doubt on the objectivity and reliability of the Logical Closeness scores.
4. The Acyclicity Assumption restricts the framework to simple forward derivation, excluding crucial reasoning patterns like planning, iterative refinement, or proof by contradiction, thereby limiting its general applicability.

**Questions:**

1. Can you demonstrate the utility of PRR/AUC by heuristically parsing DAGs from unconstrained, free-form CoT outputs (without the DAG-MATH template) on a subset of problems? If the metric collapses here, it confirms the dependency on the template is too strong.
2. Please elaborate on the relationship between DAG-MATH's diagnostic insights and existing MCTS/Tree-of-Thought techniques. How can the PRR/AUC scores be used to guide the search policies or reward functions in such systems?
3. Given the reliance on LLM-generated ground truth, what specific human review or verification process was applied to the 2,894 gold-standard DAGs to ensure their canonical logical structure? What were the human agreement statistics on the logical decomposition?

---

> ### Author Response · Authors · 2025-11-21
> **Response to Reviewer wXKz**
>
> We deeply apprecaite reviewer's effort and constructive comments on this work.
>
> First, we invite reveiwer to Common Reply 1 for detailed discussions on:
> - *applicability of PRR/AUC metric to non-formatted CoT*
> - *concern on circularity of DAG-MATH evaluation*
>
> Common Reply 2 for:
> - *concern on reliability of benchmark and human evaluation results*
>
> and Common Reply 3 for discussions on:
> - *scalability of DAG-MATH: rationale of acyclicity, applicability to reasoning patterns like planning, iterative refinement, or proof by contradiction*
>
> Next, we address reveiwer's specific concerns as below:
>
> **Q1** *relationship between DAG-MATH's diagnostic insights and existing MCTS/Tree-of-Thought techniques. How to improve such systems?*
>
> **A1** DAG-MATH differs from them on the target and the search strategies.
>
> **Graph-Level Differences:**
>
> MCTS/ToT models one solution trajectory as a **linear** chain and peform branching at each node iteratively to conduct tree search. Such linear dependency relationship cannot capture essential reasoning patterns such as shared sub-derivations and aggregations. In contrast, DAG-MATH can naturally handle flexible dependency structure.
>
> **Diagnostic Insights into Search Limitations:**
>
> 1. In mathematical reasoning, MCTS will first derive a lemma once then keep exploring alternative derivations of the same lemma, which generates redundant sub-DAGs that never merge.
> 2. A ToT is a complete k-ary tree (k is the search branching width) whose logical closure rate decays exponentially in k and depth of problem. It may score high on PASS@k but quite low on PRR.
>
> Our framework reavals such search behaviors mismatch reasoning objectives quantitatively. Table 1 in our paper shows that for identical problems, Incorrect trajectories exhibit 2–3× higher maximum out-degrees than Perfect ones, indicating that failed search explodes into speculative branching rather than focused derivation. This incoherence also imposes an extra expert-level verification burden on users.
>
> For *how to improve such systems*, we invite reveiwer to Common Reply 4 for a detailed discussions.
>
> ---
>
> We hope our response has addressed the reviewer's concerns, and we are happy to provide any further clarification if needed.

---

> ### Author Response · Authors · 2025-11-26
>
> Dear Reviewer wXKz,
>
> As the discussion period will close in one week, we kindly invite you to take a moment to confirm whether our response adequately addresses your concerns. Please let us know if any additional explanation would be helpful. Thank you very much for taking the time to review our response.
>
> Kind Regards
>
> Authors

---

> > ### Comment · Reviewer_wXKz · 2025-11-26
> >
> > Thank you for your reply. My concerns are generally resolved so I am happy to raise the score!

---

> > > ### Author Response · Authors · 2025-11-26
> > >
> > > Dear Reviewer wXKz,
> > >
> > > Thanks for your positive support. We will include your suggestions in our final version.
> > >
> > > Kind Regards
> > >
> > > Authors

---

### Author Response · Authors · 2025-11-21
**Common Reply 5 on reliable automation of the logical closeness check**

The DAG identification of our framework is compatible to interface with autoformalization and theorem-prover via LLM.

The core idea is that each step in a CoT can be treated as a natural-language statement to be autoformalized into a formal Lean statement then proven in Lean. This process would inherently perform a canonicalization, stripping away linguistic variation and leaving a unambiguous logical object.

The autoformalization and proof verification provide a critical advantage for dependency analysis: a formally formalized and proven step must explicitly include its premises from the available hypotheses and previous results. In this formalized setting, the question of "whether an edge should connect two nodes" is not a matter of subjective interpretation but of **syntactic necessity**, similar to the computational graph of programming language.

Therefore, autoformalization can provide the path to reliable automation of logical closeness. While high-accuracy autoformalization remains a challenging research frontier (e.g., Goedel-Prover-V2, Kimina-Prover, [1]) and was out of our compute budget, our DAG-MATH framework established a foundation to leverage its future success.

Alternatively, future research could leverage the DAG-MATH framework to map token-level probabilities to edge-level confidence scores, investigating whether models are well-calibrated not just on final answers, but on the intermediate logical leaps required to reach them.

**Reference:**

[1] Liu, Qi, et al. "Bootstrapping Hierarchical Autoregressive Formal Reasoner with Chain-of-Proxy-Autoformalization." NeurIPS 2025.

---

### Author Response · Authors · 2025-11-21
**Common Reply 4 on practical paths to improve LLM reasoning**

Our proposed metrics would provide a principled foundation for future work aimed at improving both search-based inference and LLM's post-training.

**1. Guiding Search Policies (MCTS/ToT):**
Our framework suggests that **logical closeness** can serve as a reward to guide graph-based search algorithms like MCTS or Tree-of-Thoughts. Instead of relying solely on random branching or value function, search policies could **prioritize reasoning paths that maintain high logical coherence**. For instance, by favoring branches that close open dependencies or by pruning highly speculative trajectories early. This offers a promising avenue to make search more efficient and goal-directed, steering it toward not just correct but also well-structured solutions.

**2. Informing Reward Design for RL:**
The AUC curves inspire a curriculum learning strategy for training reasoning models via RL. One could progressively increase the required level of logical closeness during training, starting from a regime that rewards basic correctness and gradually guiding the model toward the "sweet spot" of perfect reasoning identified by our framework.

We have included this dicussion in **Appendix J** of the updated version and leave the experimental improvement as future work.

---

### Author Response · Authors · 2025-11-21
**Common Reply 3 on the acyclicity assumption**

The acyclic assumption is fair to charecterize thinking and non-thinking models' final outputs. The LLM may reason imperfectly during its thinking process, e.g., dead-ends, iterative refinement, and backtracking, but is expected to output only the finalized, perfect, correct reasoning results to the user.
Besides, this assumption is naturally examplified by certain cases.

1. **Rationale of Acyclicity:** A mathematical derivation/proof is a finite sequence of well-formed formulas where each formula is either an axiom or is derived from earlier formulas by a rule of inference. If we treat each formula occurrence as a node and draw a directed edge from each premise inferred from it, the dependency graph we obtained is indeed a directed acyclic graph. This is also supported by formal finitary system such as Zermelo–Fraenkel set theory or type theory. Strategies like *proof-by-contradiction* are rules of inference (e.g., `¬P → ⊥ ⊢ P`), which creates a clear, acyclic dependency: the conclusion `P` depends on the sub-derivation that starts with the assumption `¬P` and ends with a contradiction `⊥`. This is not a cycle but a conditional branch that is closed, which can be covered by DAG-MATH.

2. **Compatibility with Planning:** A high-level plan can be viewed as a sketch of the solution's structure and seamlessly integrated as an initial part of the DAG-MATH, which is fully compatible with our framework.

We have updated the discussion for Assumption 1 in **Appendix A.4** of the updated version.

---

### Author Response · Authors · 2025-11-21
**Common Reply 2 on Human evaluation of gold-standard benchmark**

We thank the reviewers for raising the critical issue of potential circularity and the need for rigorous human validation of our gold-standard DAGs.

To address this, we have conducted a rigorous **Human Evaluation** (detailed in the newly added **Appendix E**) on a random sample of 50 problems. Crucially, to eliminate the subjectivity on dependency judgement in human reviews, we designed an **Algorithmic Human Evaluation Pipeline**. This approach treats the logical structure of a solution as isomorphic to a computational graph:

1.  **Methodology:** Human manually maps every natural language step of the sampled DAGs into executable **SymPy code**.
2.  **Verification:** "Dependencies" were rigorously defined as the flow of variables between lines of code. If `step_B` mathematically requires `step_A`, the code for `step_B` must reference the variable `step_A`.
3.  **Results:**
    *   **Execution Correctness:** 100% (50/50) of the stepwise conclusions and final answers are correct.
    *   **Dependency Accuracy:** 98% (49/50) of the DAGs perfectly matched the human-coded computational graph. The single anomaly was a weak narrative link rather than a logical failure.

This evaluation demonstrates that while LLMs were used to scale the construction of the benchmark, the resulting logical structures are still reliable under our strict verification desgin in **Appendix E**. Human evaluation confirms the high reliability of our DAG-MATH benchmark as a gold standard for further use such as high-quality data for post-training.

---

### Author Response · Authors · 2025-11-21
**Common Reply 1 on robustness, statistical significance, generalizability of DAG-MATH**

We performed extensive ablation studies (**Appendices K**) to show that our framework is robust to prompt variation and model's inductive bias (non-circularity). Also, we condcuct statistical tests to verify the significance of our finding claim and show the generalizability of DAG-MATH to natural CoT.

*   **Resilience to Prompting and Model Bias (Appendices K.2-4):** We confirmed that our metric and framework are robust to prompt & model variations.
    * **Prompt Perturbation:** Using Two One-Sided Tests (TOST) in **Appendix K.2**, we establish statistical equivalence in DAG structures generated by reformatted (tables vs. bullets) or rephrased prompts comparing to our originial one.
    * **Cross-Model DAG Parsing:** **Appendix K.3** uses multiple LLMs from different families to re-parse DAG structures of formatted CoT (also natural CoT) and clearly show that there are **no circularity** in DAG-MATH evaluation. The metrics remain highly close (from 24.9% to 25.5%) and nearly identical AUC trends. Furthermore, we use TOST to show statistical equivalence of DAG strcutres.
    * **Cross-Family Few-shot Demonstrations:** **Appendix K.4** addresses concerns about model-specific inductive bias; we show that using cross-family few-shot demonstrations (e.g., prompting GPT-4.1 with Gemini-generated trajectories) yields consistent relative rankings and metric behavior, also achieve statistical equivalence on DAG structures via TOST, proving the evaluation is not dependent on matching the generator’s own reasoning style.

*   **Statistical Significance of the Reasoning Gap (Appendix K.1):** We verified that the observed disparity between final-answer accuracy (PASS@1) and reasoning rigor (PRR) is not due to sampling noise. By scaling the evaluation to 128 samples per problem, we computed tight 95% confidence intervals for this gap. The results consistently exclude zero with non-trivial values (from 9.0% to 36.3%), statistically confirming that the gap between finding the correct answer and adhering to a valid deductive path is a substantial and genuine phenomenon in current LLMs.

*   **Generalizability to Natural CoT (Appendix K.3):** Our evaluation metrics (PRR and AUC) are not limited to the structured DAG-MATH format. By parsing free-form **Natural CoT** via both **internal & external models**, we found that logical closeness remains **consistent** across different DAG parsings. The metrics successfully capture fundamental reasoning patterns in natural text, achieving AUC scores comparable to the structured format, which validates the framework’s utility beyond our specific benchmark format.

---

### Author Response · Authors · 2025-11-21
**Revision summary**

We deeply thank for the reviewers' insightful comments and suggestions. We address the common concerns and then respond each reviewer's specific question/weakness point-by-point. We have updated the paper (highlight in red) to include these insightful suggestions:

**Major:**

\+ **Appendix E** Human evaluation of gold-standard DAG-MATH benchmark (reviewers `wXKz`, `utCQ`, `GdSj`)

\+ **Appendix K** Extended experimental studies on:
- Statistical significance test of reasoning gap with extended sample size (reviewer `utCQ`)
- Ablation studies on the sensitivity of prompt via perturbation (reviewer `utCQ`)
- Generalization of DAG-MATH format to natural CoT, and ablation studies on non-circularity of DAG-MATH evaluation (reviewers `wXKz`, `61B1`)
- Ablation studies on robustness of evaluation using cross-family few-show demonstrations (reviewer `61B1`)

We summarize the key findings from above studies into one new discussion section in **Section 5.2**.

**Minor:**

\+ **Appendix A.4** Add discussion on rationale of acyclicity assumption (reviewers `wXKz`, `61B1`).

\+ **Appendix D** Add more details of validation procedure in benchmark construction and filter statistics (reviewer `GdSj`).

\+ **Appendix J** Add discussions on future improvements from our evaluation insights (reviewers `wXKz`, `utCQ`).

---

### Author Response · Authors · 2025-11-29
**Rebuttal Summary for Area Chair**

Dear AC,

Thank you for your time and effort in handling our submission. Below is a brief summary of the reviewers’ assessments and our clarifications.

Our work introduces a novel **DAG-based framework** to model and evaluate step-level CoT via **new metrics**. The proposed DAG-MATH benchmark, together with extensive empirical analysis, provides a principled way to evaluate reasoning ability and to diagnose when, why, and how reasoning processes fail.
This work is recongized as **fundamental** (highlighted by reviewers `wXKz`, `utCQ`, and `GdSj`).

In the rebuttal, we addressed all reviewer concerns (see **Revision Summary**), mainly including:
- our framework is reliable and robust to prompt variation and model's inductive bias.
- the rationality of the acyclicity assumption: valid for thinking models with self-correction and planning
- the possibility of imrpoving LLM reasoning

Reviewer `wXKz` has already acknowledged the clarifications and increased their score from 4 to 6 (as recorded on Nov. 26, prior to the OpenReview data leak issue). All reviewers expressed positive support overall.

We hope this summary assists your decision-making.

Best,

Authors

---

### Meta-Review · Area_Chair_Qn2x · 2026-01-06

**Summary:**

This paper proposes a directed acyclic graph (DAG)-based framework for modeling and evaluating CoT trajectories. Within the framework, this paper introduces a metric, logical closeness, to quantify the logical consistency within the models' reasoning process. The paper thus proposes the DAG-MATH format for CoT and constructs a benchmark dataset with 2,894 LLM-generated gold-standard DAG-MATH DAGs. The experimental results indicate significant differences in reasoning fidelity among LLM families.

**Reviewer Concerns:**

Addressed concerns:
* Circularity in benchmark & LLM-as-judge reliability (Reviewers utCQ, GdSj): With human evaluation (Appendix E) and cross-model parsing consistency (Appendix K.3).
* Statistical significance tests for the differences between PASS@1 and PRR (Reviewer utCQ): Confirmed by extending the sample size from 32 to 128 with confidence intervals in Appendix K.1.
* Sensitivity of prompting strategy (Reviewer utCQ): Two One-Sided Tests (TOST) in Appendix K.2.
* Qualitative analysis/case study (Reviewer 61B1): Highlighted case study and analysis in Appendix B in the original manuscript.
* Empirical results for thinking LLMs (Reviewer 61B1): Added experiments in Section 5.2.
* Clarification on Figure 4 (GdSj): Caption clarified in the revised manuscript.

Partially addressed concerns:
* Theoretical justification of DAG formalization: The explanation remains more operational than foundational.
* Reliable automation of the logical closeness check: Relies on future autoformalization
* Scalability of DAG-MATH: May be limited by the acyclic assumption.

**Reviewer Scores:**

The reviewers have a high potential of raising their scores, especially Reviewer wXKz, given the detailed and comprehensive author response.

---

### Decision · Program_Chairs · 2026-01-26

Accept (Poster)